

**Aerosol-Driven Precipitation Modification: Spatiotemporal Heterogeneity in**
**Precipitation Microphysics and Vertical Structures over China's Megacity**
**Clusters**
Heyuan Peng[1], Xiong Hu[2*], Weihua Ai[1*], Zhen Li[1], Shensen Hu[1], Junqi Qiao[1],
Xianbin Zhao[1]
[1]*College of Meteorology and Oceanography, National University of Defense*
*Technology, Changsha, China*
[2]*Basic Education College, National University of Defense Technology, Changsha,*
*China*
*\*Corresponding authors: X. Hu (huxiong18@nudt.edu.cn) and W. Ai*
*(aiweihua@nudt.edu.cn)*
**ABSTRACT**
As crucial atmospheric components, aerosols influence precipitation through
complex microphysical mechanisms and exhibit spatiotemporal heterogeneity. This
study investigates aerosol effects on precipitation vertical structures and
microphysical characteristics across four Chinese urban clusters (the
Beijing–Tianjin–Hebei (BTH), Yangtze River Delta (YRD), Yangtze River Middle
Reaches (YRM), and Pearl River Delta (PRD)), including sensitivities to
meteorological factors. Initially, the principal findings elucidate three fundamental
attributes of precipitation differences: regional disparities surpass seasonal variations
in magnitude; heightened aerosol concentrations mitigate regional precipitation



discrepancies, particularly during the spring and summer seasons; convective
precipitation exhibits greater regional and seasonal variability than stratiform
precipitation. Furthermore, the findings indicate that aerosols exert an influence on
precipitation through microphysical processes, encompassing the growth via
condensation on cloud condensation nuclei, coalescence growth, semi-direct effect,
and moisture competition. These phenomena exhibit distinct variations that are
influenced by spatial and temporal factors, as well as the particular type of aerosols
present. Specifically, convective precipitation in the BTH region is dominated by the
semi–direct effect of dust aerosols, whereas the YRD and PRD are more influenced
by hygroscopic sea salt aerosols and the YRM by fine aerosol particles. Furthermore,
RH promotes condensation and coalescence processes by replenishing water vapor,
particularly under low aerosol loading. However, CAPE plays a dual role: it enhances
precipitation by intensifying cloud development and suppresses it through particle
break-up driven by dynamics. The present study elucidates the mechanisms of
spatio–temporal modulation underlying aerosol–precipitation interactions, offering a
scientific foundation for the refinement of climate models within urban
agglomerations.
**Key words:** GPM DPR, MERRA–2, Aerosols, Precipitation Structure, Urban
Agglomerations



## 1. Introduction

Aerosols modulate clouds and precipitation primarily through Aerosol–Radiation
Interactions (ARI) and Aerosol–Cloud Interactions (ACI). These mechanisms affect
the intensity, frequency, and spatiotemporal distribution of precipitation (Rosenfeld et
al., 2008). These processes involve complex multiscale, multi–factor coupling effects
with profound implications for regional hydrological cycles, extreme weather events,
and climate systems (Li et al., 2016, 2019; Ramanathan et al., 2001). Therefore, one
of the most important problems facing atmospheric research is the clarification of
aerosol–driven precipitation mechanisms (IPCC, 2013; IPCC, 2021). In this
framework, the ACI describes the mechanism by which aerosols function as ice nuclei
(IN) and cloud condensation nuclei (CCN), modifying cloud microphysical processes
to indirectly modify the type and distribution of precipitation intensity (Gettelman,
2015). These include cloud droplet spectrum distribution, phase transition efficiency,
and precipitation formation pathways (Xie et al., 2013). Known as the Twomey Effect
(the First Indirect Effect), higher aerosol concentrations increase the number of cloud
droplets, while decreasing their effective radius ($r_e$) and increasing cloud albedo
(Twomey, 1974). In addition, the Cloud Lifetime Effect (Second Indirect Effect),
aerosol-induced reduction in $r_e$, suppresses precipitation initiation, whereas
prolonging cloud lifetime (Albrecht, 1989). The Semi-Direct Effect is another way in
which absorptive aerosols can shorten cloud lifetime by heating the atmosphere
through the absorption of shortwave radiation, which accelerates droplet evaporation
(Ackerman et al., 2000; Huang et al., 2014). There are still many unknowns





surrounding the measurement of aerosol impacts on precipitation, even though the
primary mechanisms of aerosol–precipitation interactions are documented. This is due
to the diversity and highly nonlinear characteristics of aerosol–precipitation responses
(Chang et al., 2015), which are jointly regulated by aerosol concentration, type,
vertical distribution, and local meteorological conditions (Fan et al., 2007; Storer et al.,
2010), leading to pronounced regional variations (Xiao et al., 2025). Furthermore,
external synoptic conditions modulate the ACI process (Chen et al., 2025; Sun et al.,
2023; Zhao et al., 2024).

Significant research in recent years has focused on aerosol–induced modifications

of precipitation structures in key regions of China. Major urban agglomerations,
Beijing–Tianjin–Hebei (BTH), Yangtze River Delta (YRD), Pearl River Delta (PRD),
and Yangtze River Middle Reaches (YRM)–represent China's most economically
vibrant and densely populated areas while also experiencing severe aerosol pollution
(Guo et al., 2018; Sun and Zhao, 2021; Zhao et al., 2025), thus providing critical entry
points for investigating regional manifestations of aerosol effects. Although previous
research has been conducted on precipitation patterns in more general areas such as
the North China Plain (Sun et al., 2023), South China (Chen et al., 2025), and East
China (Wen et al., 2023), analysis of specific seasons or precipitation types is
frequently limited without considering meteorological drivers. Moreover, numerical
models show substantial instability in precipitation capture (Zhang et al., 2024), and
simulation capabilities exhibit inherent asymmetries (Snively and Gallus, 2014). In
summary, methodological divergences and data source variations across studies have



yielded divergent conclusions with persistent controversies, precluding robust
cross–regional comparisons of aerosol impacts on precipitation structures. Therefore,
it is essential to develop consistent techniques for the collection and interpretation of
aerosol–precipitation data.

The Global Precipitation Measurement (GPM) mission extends and advances the

Tropical Rainfall Measuring Mission (TRMM). Compared with TRMM's
single–frequency Precipitation Radar (PR), the Dual–frequency Precipitation Radar
(DPR) onboard the GPM core observatory demonstrates higher sensitivity and
provides more accurate three–dimensional precipitation structure. This increase
markedly enhances precipitation detection capabilities at mid-to high latitudes (Hou et
al., 2014). Furthermore, comparisons between GPM DPR precipitation data and
observations from ground–based radars and meteorological stations (Chandrasekar
and Le, 2015; Lasser et al., 2019; Sun et al., 2020) validated the substantial agreement
across all three platforms. Moreover, a robust concordance in surface precipitation
patterns and brilliant band height was noted between DPR data and the
high–resolution NICAM 3.5 km model (Kotsuki et al., 2023), hence reinforcing data
dependability.

Additionally, Modern–Era Retrospective Analysis for Research and Applications

Version 2 (MERRA–2) significantly improves the accuracy of aerosol vertical
distributions and optical properties through assimilation of multi–source satellite and
ground–based observations (Buchard et al., 2017; Chang et al., 2015). Building on the
reliable precipitation data from GPM DPR, researchers analyzed aerosol impacts on





precipitation vertical structure, microphysical characteristics, and extreme
hydrometeorological events using integrated MERRA–2 aerosol and DPR
precipitation datasets (Ji and Tian, 2024; Jiang et al., 2023; Sun et al., 2022).
Furthermore, compared with the ECMWF Re–Analysis–Interim (ERA–Interim), the
European Centre for Medium–Range Weather Forecasts Reanalysis Version 5
(ERA–5) offers significantly improved spatiotemporal resolution, yielding superior
environmental parameters(Zhao et al., 2021). This enhancement facilitates its
widespread utilization in research investigating the impact of aerosol on precipitation
structure(Dong et al., 2018; Guo et al., 2018; Pravia-Sarabia et al., 2023).
Peng et al. (2025) conducted a focused investigation into the effects of fine and
coarse aerosols on summer precipitation processes within the YRD region. The results
indicated that coarse aerosols suppress convective precipitation by competing for
moisture, whereas fine aerosols enhance precipitation by forming small droplet
clusters with condensational and coalescence growth. However, precipitation
characteristics vary significantly across different regions and seasons, which can be
attributed to differences in aerosol types, concentrations, and meteorological
conditions. Therefore, this study integrates precipitation, aerosol, and environmental
data from four major urban agglomerations (the BTH, YRD, YRM, and PRD)
between 2014 and 2023. A multi–source DPR–MERRA–2–ERA5 dataset was
constructed to systematically analyze the impact of aerosol on precipitation properties
and microphysical processes. In addition, the interactions between aerosol and
precipitation structures are further investigated under varying thermodynamic and





dynamic conditions. This unified methodology facilitates a comprehensive
examination of aerosol effects on the precipitation structure and cloud microphysics
across China's major urban agglomerations, enabling cross–regional comparative
assessments.
The remainder of this paper is organized as follows. Section 2 introduces the data
and methods. Section 3 examines aerosol influences on precipitation structure and
properties. Section 4 conducts an analysis of aerosol effects on the microphysical
processes of precipitation. Section 5 investigates meteorological effects on
aerosol-precipitation interactions. Section 6 summarizes the conclusions of this study
and Section 7 discusses the limitations and shortcomings of this research.
**2. Data and Methods**
*2.1 Study area*
The YRM urban cluster (Fig. 1a) is situated between 26°N–32.5°N and
110.5°E–118.3°E, inside a humid subtropical monsoon region characterized by
concentrated summer precipitation. The BTH region (36°N–41.6°N, 113.5°E–119.9°E;
Fig. 1b) exhibits a temperate semi–humid continental monsoon climate, with summer
comprising over 67% of the annual precipitation and spring characterized by
numerous dust events (Zhai et al., 2022). The PRD cluster (21.7°N–23.8°N,
112°E–115.4°E; Fig. 1c) exhibits a South Asian tropical marine monsoon climate,
characterized by the 85% of annual precipitation occurring from April to September,
frequently exacerbated by typhoons (Guo et al., 2018). Dominated by the Taihu Plain



(27.9°N–33.3°N, 117.5°E–122.7°E; Fig. 1d), the YRD exhibits a humid subtropical
monsoon climate characterized by concentrated spring–summer precipitation,
including prolonged June–July Meiyu–front rainfall (Liu et al., 2017). Fig. 1 shows
the spatial distribution of all four urban agglomerations.

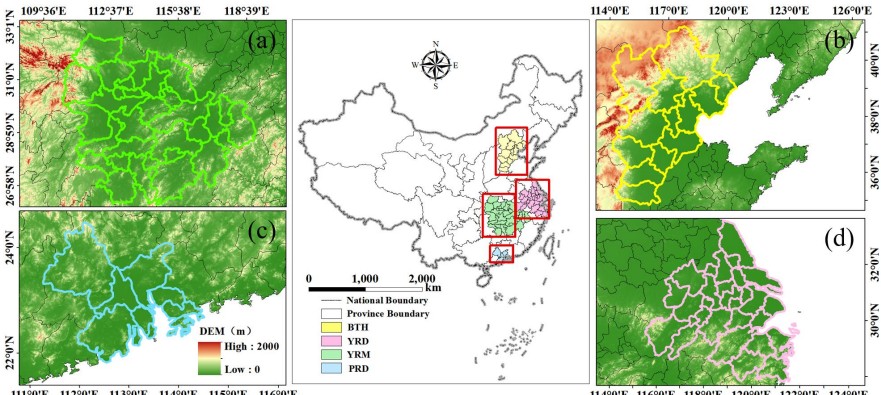

**Fig. 1.** Geographical location and elevation map of the Yangtze River Middle
Reaches (a), Beijing–Tianjin–Hebei (b), Pearl River Delta (c), and Yangtze River
Delta (d) urban agglomeration (source: GS(2024) 0650). Publisher's remark: please
note that the above figure contains disputed territories.
***2.2 GPM DPR Precipitation Data***
Mounted on the core satellite, the DPR transmits at the Kuband (13.6 GHz) and
Kaband (35.5 GHz) frequencies, achieving a nadir horizontal resolution of 5 km to
detect three–dimensional precipitation structures from the surface to an altitude of 22
km (Hou et al., 2014). This study utilized the GPM Level 2 DPR (2ADPR) standard
product, Version 07, which employs two unique antenna scanning modes: the
High–Sensitivity Scan (HS) and the Full Scan (FS). A significant alteration in Version
07, compared to Version 06, entails a shift in the KaPR scanning pattern from the



inner swath to the outer swath configuration. This change aligns the KaPR scanning
mode with that of KuPR.

Given the relatively low frequency of winter precipitation events and frequent

occurrence of solid precipitation in northern regions, this study focuses on
precipitation data during spring (March–May), summer (June–August), and autumn
(September–November) from 2014 to 2023. The parameters analyzed in this study
include: The near–surface Rain Rate (nsRR), Rain Rate (RR), Storm Top Height
(STH), Liquid Water Path (LWP), Ice Water Path (IWP), DSD, and radar reflectivity
factor ($Z_e$). The DSD includes two parameters: mass–weighted diameter ($D_m$) and
normalized DSD intercepts ($N_w$).
***2.3 MERRA–2 Aerosol Data***

This study utilized the MERRA–2 atmospheric reanalysis dataset, updated in

2017 and released by NASA's Global Modeling and Assimilation Office (GMAO).
By assimilating multi–source observations with numerical modeling techniques, this
dataset characterizes the column mass concentrations of five aerosol types: dust (DU),
sea salt (SS), sulfate (SO4), black carbon (BC), and organic carbon (OC), and their
corresponding AOD. The data feature a global spatial coverage at a horizontal grid
resolution of 0.625°×0.5° (longitude×latitude), with temporal products available at
hourly intervals.

It is noteworthy that the aerosol data matched are those prior to precipitation

events.



***2.4 ERA–5 Data***
Environmental data for this study were acquired from the ERA5 reanalysis
dataset. Through the coupled assimilation of multi–source satellite observations,
ground–based measurements, and numerical forecasting systems, this product
provides multidimensional climate parameters that span the surface–to–stratopause
column (Hersbach et al., 2020). This investigation utilizes two key parameters:
Relative Humidity (RH) at 850hPa and Convective Available Potential Energy
(CAPE).
***2.5 Classification Methods***
Prior to data analysis, the DPR, MERRA–2, and ERA5 datasets were subjected to
spatiotemporal matching using the best–proximity method. Subsequently,
precipitation pixels were screened using the connectivity method (Hu et al., 2022),
applying a minimum threshold of four contiguous pixels to define valid precipitation
systems. This study categorized precipitation into stratiform and convective types
based on the 2ADPR classification criteria, excluding shallow convection events from
convective precipitation (Liu and Zipser, 2015). Aerosol classification followed the
total AOD thresholds: Low AOD, [0, 0.3); Medium AOD, [0.3, 0.6); and High AOD,
[0.6, ~). In terms of aerosol classification, BCA, OCA, and $SO_{4A}$ were categorized as
fine aerosol particles, whereas SSA and DUA were classified as coarse aerosol
particles.
Fig. S1 illustrates the cumulative distribution functions (CDFs) of the RH and
CAPE throughout the four urban agglomerations for meteorological conditioning.





Given the higher similarity in RH distributions among the YRD, YRM, and PRD
versus distinct BTH characteristics, the YRD–YRM–PRD RH data were unified (red
dashed line; Fig. S1a–c). Conversely, the BTH–YRD–YRM exhibited comparable
CAPE distributions, and the PRD showed significantly higher values. Thus, the
BTH–YRD–YRM CAPE data were combined (red dashed line; Fig. S1d–f). To
balance methodological consistency with regional specificity, the classification
strategy implemented distinct groupings: RH was classified separately for the BTH
region, whereas the YRD, YRM, and PRD shared a unified RH classification.
Similarly, CAPE maintained independent classification for the PRD, whereas BTH,
YRD, and YRM employed combined CAPE classification, as visualized by the red
dashed lines in Fig. S1. Moreover, to prevent feature ambiguity from adjacent
samples, three percentile tiers were defined using the CDFs thresholds: low
(0%–30%), medium (35%–65%), and high (70%–100%), with 30–35% and 65–70%
as buffer zones to avoid adjacent–sample ambiguity.
***2.6 Normalized difference calculation***
In order to quantify regional and seasonal differences in precipitation parameters,
the BTH region and spring season were set as the benchmark for normalizing
variations. The fractional changes (DIFF$_{region}$, in %) for each parameter in the YRD,
YRM, and PRD regions relative to BTH were calculated as follows:

$$\mathrm{DIFF}_{region} = \frac{\mathrm{X}_{region} - \mathrm{X}_{BTH}}{\mathrm{X}_{BTH}} * 100\%$$      (1)

Beyond regional differences, the fractional seasonal changes (DIFF$_{season}$, in %)
for precipitation parameters were calculated as:





$$\text{DIFF}_{\text{season}} = \frac{X_{\text{season}} - X_{\text{spring}}}{X_{\text{spring}}} * 100\% \tag{2}$$


where $\text{DIFF}_{\text{region}}$ represents the normalized differences for the YRD, YRM, and PRD
relative to BTH, respectively; $\text{DIFF}_{\text{season}}$ denotes the normalized seasonal differences
when comparing summer and autumn to spring. $X_{\text{BTH}}$ and $X_{\text{spring}}$ represent the
reference value of the target precipitation parameter in the BTH and spring,
respectively. $X_{\text{region}}$ denotes the precipitation parameter values for the YRD, YRM,
and PRD regions, respectively, and $X_{\text{season}}$ represents the precipitation parameter
values in the seasons being compared to spring.
**3 Influence of aerosols on precipitation structure and properties**
*3.1 Correlation changes of precipitation parameters with aerosols*
To investigate aerosol impacts on convective and stratiform precipitation
characteristics across the four urban agglomerations, five precipitation parameters
were selected: nsRR, STH, LWP, IWP, and the precipitation efficiency index (PEI). A
correlation heat map of the convective precipitation between AOD and these
parameters is shown in Fig. 2. The PEI is a crucial metric for measuring the efficiency
of precipitation formation in clouds, indicating the effectiveness of converting cloud
water into precipitation. Higher PEI values indicate enhanced precipitation efficiency,
which is characterized by a greater conversion of cloud water into rainfall. Following
Hu et al. (2022) and scaled by 1000 for enhanced readability, PEI is defined as:

$$\text{PEI} = \frac{\text{nsRR}}{\text{CWP}} = \frac{\text{nsRR}}{(\text{LWP} + \text{IWP})} * 1000 \tag{3}$$






In the resulting heat map of convective precipitation (Fig. 2), individual cells
exhibit Spearman correlation coefficients that quantify the relationship between AOD
and key meteorological parameters. For spring convective precipitation (Fig. 2a–c),
the BTH exhibits significant negative correlations with the STH and IWP under low
aerosol loading, whereas showing a significant positive correlation with PEI. The
YRD displays patterns similar to those of the BTH: negative correlations at low
aerosol loading shift to positive correlations with precipitation parameters as aerosol
loading increases. In contrast, both the YRM and PRD show consistently positive
correlations under low aerosol loading (Fig. 2a). However, the PRD demonstrates
pronounced negative correlations at moderate loading, whereas the YRM maintains
positive correlations at high aerosol loading. Summer (Fig. 2d–g): Under low (high)
AOD conditions, consistent positive (negative) correlations prevail across all study
regions (Fig. 2d). At moderate aerosol loading levels, the PRD shifts to a negative
correlation, whereas the other three regions retain a positive correlation (Fig. 2e).
Autumn (Fig. 2g–i): the YRD exhibits pronounced negative correlations under
low–to–moderate AOD thresholds. Conversely, the BTH and YRD (PRD)
demonstrate positive (negative) correlations under high AOD levels. Overall, the
precipitation under varying aerosol loading exhibits pronounced seasonal and regional
disparities, demonstrating nonlinear characteristics in their relationships.

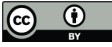

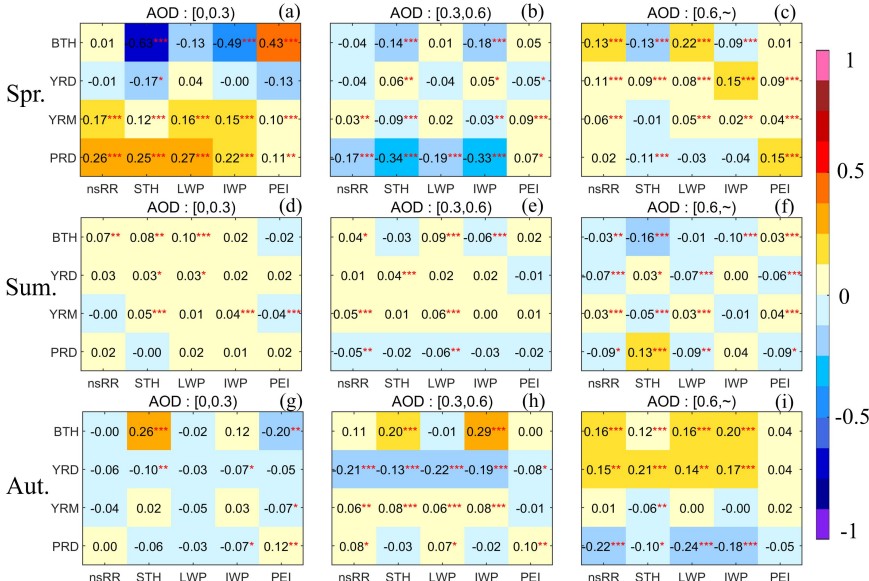

**Fig. 2.** Spearman correlation coefficients between AOD and precipitation parameters for convective precipitation across regions and seasons under the three AOD regimes. Color gradients (from yellow to blue) encode the correlation strength and direction, and asterisks denote statistical significance (*: *p*<0.05, **: *p*<0.01, ***: *p*<0.001).

Spearman correlation coefficients are computed to characterize the precipitation parameters in stratiform precipitation (Fig. S3), similar to that of convective precipitation (Fig. 2). Overall, the PRD exhibits the strongest similarity between the stratiform and convective precipitation parameters in the correlation with AOD. In contrast, the BTH, YRD, and YRM resemble convective precipitation characteristics under moderate to high aerosol loading, but show reduced similarity under low aerosol loading, particularly in the BTH.

***3.2 Changes in the structural characteristics of precipitation parameters associated with aerosols***





Fig. 3 illustrates the seasonal mean values of the convective precipitation
parameters for nsRR, STH, LWP, IWP, and PEI across AOD intervals. During spring
(Fig. 3a–e), the BTH and YRD regions exhibit similar responses, with nsRR, STH,
LWP, and PEI demonstrating persistent enhancement as the aerosol burden increases.
In contrast, the YRM and PRD manifest nonlinear features: nsRR, LWP, and IWP
experience initial suppression before rebounding at elevated AOD levels. Summer
observations (Fig. 3f–j) reveal predominantly linear positive relationships between
aerosol loading and precipitation parameters in the BTH, YRD, and YRM regions.
However, the PRD diverges sharply, displaying inverted V–shaped responses in the
nsRR, STH, LWP, and PEI. Autumn analysis (Fig. 3k–o) reveals consistent increases
in the YRM and PRD with aerosol enhancement, whereas the BTH and YRD exhibit
a pattern of initial decline followed by a subsequent increase.
The regional comparisons indicate that the differences of the precipitation
parameters are mitigated under moderate to high aerosol loading conditions (Table 1).
For instance, at low aerosol loading, the fractional changes in spring nsRR reach
$DIFF_{PRD}$=613% and $DIFF_{YRM}$=247.33%, whereas high aerosol loading reduces these
to $DIFF_{PRD}$=155% and $DIFF_{YRM}$=49.6% (Table 1). These results highlight that
increasing aerosol loading can moderately alleviate regional disparities in
precipitation characteristics. Furthermore, the BTH exhibits notably higher IWP and
PEI values, indicating enhanced ice-phase processes and superior precipitation
conversion efficiency, as evidenced by the negative $DIFF_{YRD}$, $DIFF_{YRM}$, and $DIFF_{PRD}$
values.



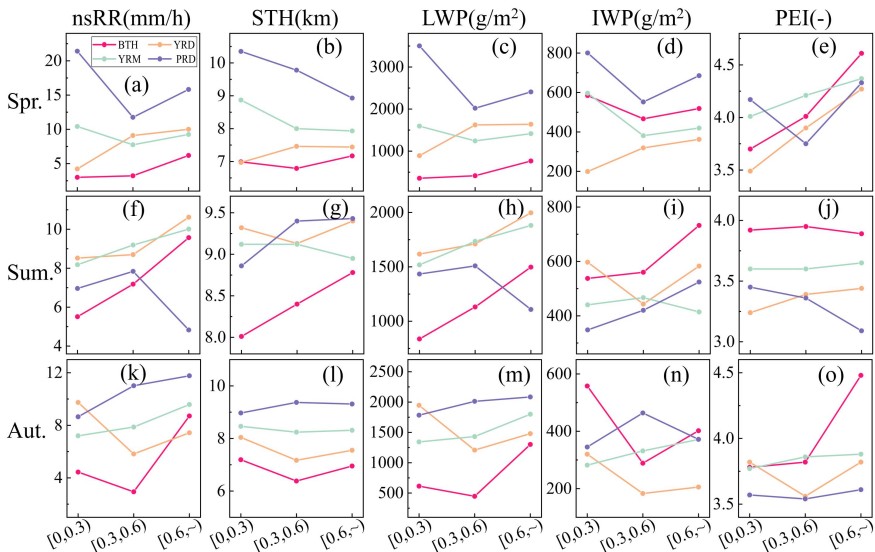

**Fig. 3.** Average point line plots of nsRR, STH, LWP, IWP, and PEI under three AOD

conditions for convective precipitation across the four regions and seasons. Each

subplot employs color–coding ( BTH–red, YRD–yellow, YRM–green, and

PRD–purple) with the x–axis denoting AOD bins ([0,0.3), [0.3,0.6), [0.6, ~)) and the

y–axis representing parameter magnitudes.



**Table 1.** Normalized regional differences in convective precipitation during the
spring season. Units: nsRR (mm/h), STH (km), LWP (g/m$^2$), IWP (g/m$^2$).

|  | AOD | BTH | YRD DIFF$_{YRD}$ | YRM DIFF$_{YRM}$ | PRD DIFF$_{PRD}$ |
|---|---|---|---|---|---|
| nsRR | [0,0.3) | 3 | 4.23+41.00% | 10.42+247.33% | 21.4+613.33% |
| | [0.3,0.6) | 3.23 | 9.1+181.73% | 7.75+139.94% | 11.76+264.09% |
| | [0.6,~) | 6.19 | 10.01+61.71% | 9.26+49.60% | 15.82+155.57% |
| STH | [0,0.3) | 6.99 | 6.97-0.29% | 8.87+26.90% | 10.35+48.07% |
| | [0.3,0.6) | 6.79 | 7.46+9.87% | 8+17.82% | 9.78+44.04% |
| | [0.6,~) | 7.17 | 7.44+3.77% | 7.93+10.60% | 8.93+24.55% |
| LWP | [0,0.3) | 355.11 | 892.42+151.31% | 1593.58+348.76% | 3501.97+886.16% |
| | [0.3,0.6) | 414.57 | 1622.99+291.49% | 1242.87+199.80% | 2019.85+387.22% |
| | [0.6,~) | 765 | 1639.19+114.27% | 1415.35+85.01% | 2407.82+214.75% |
| IWP | [0,0.3) | 584.6 | 199.12-65.94% | 596.38+2.02% | 800.57+36.94% |
| | [0.3,0.6) | 466.54 | 319.12-31.60% | 381.24-18.28% | 552.17+18.35% |
| | [0.6,~) | 518.72 | 362.2-30.17% | 419.48-19.13% | 685.27+32.11% |
| PEI | [0,0.3) | 3.7 | 3.49-5.68% | 4.01+8.38% | 4.17+12.70% |
| | [0.3,0.6) | 4.01 | 3.9-2.74% | 4.21+4.99% | 3.75-6.48% |
| | [0.6,~) | 4.61 | 4.27-7.38% | 4.37-5.21% | 4.33-6.07% |

To further characterize the stratiform precipitation parameters (nsRR, STH, LWP,
IWP, and PEI), the seasonal mean values across aerosol loading levels are presented
in Fig. S4 through point–line plots that were formatted consistently with those in Fig.

3.

For spring (Fig. S4a–e), the BTH, YRD, and YRM exhibit continuously
increasing trends in nsRR, LWP, and IWP with increasing aerosol loading.
Conversely, the PRD shows overall decreasing trends across most precipitation
parameters. In summer (Fig. S4f–j), the BTH, YRD, and YRM demonstrate linear
increases in nsRR and LWP means as aerosol loading rises. However, the PRD
exhibits nonlinear trends: initial increases followed by decreases in nsRR, STH, LWP,
and IWP means, a pattern consistent with its convective precipitation behavior. In
autumn (Fig. S4k–o): monotonically rising trends in nsRR, LWP, and IWP are shown

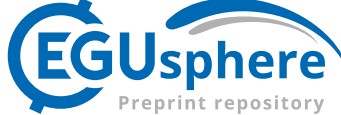

by the YRD and YRM, whereas the BTH and PRD display decreasing and then
increasing trends in nsRR, IWP, and PEI, indicating an increase in aerosol loading.
Additionally, the fractional changes in stratiform precipitation indicate that an
increase in aerosol loading moderately reduces regional disparities during regional
normalized difference comparisons, particularly in the spring and summer
(Tables .S3-5). For instance, in the PRD region during spring (Table .S3), the
DIFF$_{PRD}$ values for IWP are 208.16%, 141.06%, and 34.13%, corresponding to AOD
ranges of [0, 0.3), [0.3, 0.6), and [0.6, ~), respectively. The seasonal normalized
differences indicate significant ice-phase processes but weak liquid-phase processes
across various regions during spring.
To summarize, aerosols influence the average values of precipitation parameters,
displaying characteristics that differ across spatiotemporal scales and precipitation
types. Furthermore, precipitation parameters exhibit greater regional than seasonal
variation. In particular, within the 270 DIFF$_{region}$ samples, 41 (constituting 15.2%)
exhibited values exceeding 100%, whereas among the 240 DIFF$_{season}$ samples, only 10
(representing 4.2%) demonstrated such a phenomenon (Tables S.1-7). Additionally,
convective precipitation shows larger magnitude changes across seasons and regions
than stratiform precipitation.
***3.3 Changes in the vertical structure of precipitation associated with aerosols***
To further investigate the vertical structure of precipitation, Fig. 4 presents the
vertical distributions of the mean $Z_e$, $D_m$, $N_w$, and RR for convective precipitation
under varying AOD loadings and seasons. Overall, the mean $Z_e$, $N_w$, and RR values



generally increase with decreasing height, whereas $D_m$ exhibits an initial decrease
followed by an increase. In addition, the PRD displays a markedly higher RR than the
BTH, YRD, and YRM in spring, but lower RR in summer. This seasonal contrast may
be attributed to the abundant moisture supply during the pre–rainy season in South
China, versus significant precipitation suppression in summer caused by hygroscopic
aerosol–induced moisture competition. Furthermore, Chen et al. (2015) observed that
a moderate CAPE in the PRD summer may lead to diminished precipitation.
As shown in Fig. 5a, in spring, the RR and $Z_e$ of BTH, YRD, YRM, and PRD
have obvious similarities with the variation in AOD. The RR and $Z_e$ increase linearly
with aerosol loading in the BTH and YRD but display non–monotonic trends (initial
suppression followed by enhancement) in the YRM and PRD, consistent with Fig.
4a–e. This discrepancy may arise from the predominant influence of DUA on the
spring AOD composition of the BTH region (Fig. S1a), whereby their semi–direct
effect promotes cloud droplet evaporation, reduces moisture availability, and triggers
earlier precipitation(Sun and Zhao, 2021). Although the rising proportion of
fine–mode aerosols mitigates dust–induced precipitation suppression as AOD
increases, persistently low moisture availability in the BTH maintains precipitation
levels below those of the other regions. The YRD exhibits a more abundant water
vapor supply (Fig. S1a): Increased aerosol concentrations supply additional CCN and
IN, providing more condensation nuclei for cloud droplet formation and thereby
enhancing precipitation. With increasing AOD, the YRM transitions from
low–concentration coarse particles to high–concentration fine particles (Fig. 4c–d).





Inadequate moisture supply triggers the Twomey effect through competition for cloud
water; however, extended cloud longevity and enhanced collision–coalescence (Fig.
5a) subsequently cause precipitation to initially decline and then rise. The PRD
precipitation patterns resemble those of the YRM, although hygroscopic SSA exert a
stronger influence (Fig. S2a). At low AOD (with an SSA proportion of 26.32%),
moisture from South China's pre–rainy season and hygroscopic giant CCN derived
from sea salt promote spring precipitation growth (Guo et al., 2022). As AOD
increases,    the    SSA    contribution    declines    rapidly,    weakening    its
precipitation–promoting effect. However, with further AOD growth, the proportion of
fine (notably hygroscopic OCA) aerosols rises (Fig. S2a), supplying more effective
CCN to enhance the precipitation. Additionally, under low AOD conditions, the lower
atmosphere in the YRD (YRM) is dominated by high concentrations of smaller (larger)
particles. As aerosol loading increases, the Twomey effect emerges in the YRM,
whereas the BTH and YRD exhibit the anti–Twomey effect.
During the summer (Fig. 4e–h), average vertical profiles of RR and $Z_e$ in the BTH,
YRD, and YRM generally exhibit increasing linear trends with rising aerosol loading.
Conversely, the PRD shows an initial increase followed by a decrease, which is
consistent with the findings in Fig. 3f–j. Within the BTH, YRD, and YRM, low
proportions of hygroscopic giant CCN (SSA) mean increasing aerosols boost $N_w$ and
$D_m$, supplying more CCN and IN for cloud droplet formation. Coupled with ample
summer moisture supply and dynamic forcing, cloud droplets are transported to
higher altitudes (Fig. 3g), enhancing precipitation. During summer in the PRD, ample





moisture is derived from the Indian Ocean. With low aerosol loading, the proportion
of sea salt particles is elevated (23.83%). An increase in giant sea salt CCN loading
triggers an anti–Twomey effect, characterized by a rise in $D_m$ and a decline in $N_w$,
leading to intensified precipitation. Nevertheless, as aerosol loading escalates further,
fine–mode particles predominate (93.9%; Fig. S2b), resulting in moisture competition
becoming the principal mechanism inhibiting precipitation.
Consistent with the conclusions in Fig. 3k–o, autumn trends (Fig. 4i–l) show that
precipitation in the BTH and YRD initially decreased and then increased with
increasing aerosol loading, although the magnitude of the increase is increasing with
rising aerosol loading, though the increase magnitude is larger in the BTH than in the
YRD. Conversely, RR exhibits monotonically increasing trends in the YRM and PRD.
The underlying mechanisms are as follows. The BTH and YRD: Initial aerosol
increase elevates $N_w$ while reducing $D_m$ (Fig. 4k–l), suppressing precipitation via the
Twomey effect. However, prolonged cloud lifetime promotes further cloud
development (Fig. 3l–m). Consequently, when aerosol loading continues rising,
abundant CCN and IN become available for cloud–precipitation processes, ultimately
enhancing precipitation. The YRM and PRD: The monotonic trends stem from greater
autumn moisture availability versus the BTH and YRD (Fig. S1c), which supports
cloud droplet condensational growth (Fig. 4k–l) and enhances collision–coalescence
(Fig. 5a), collectively facilitating precipitation.



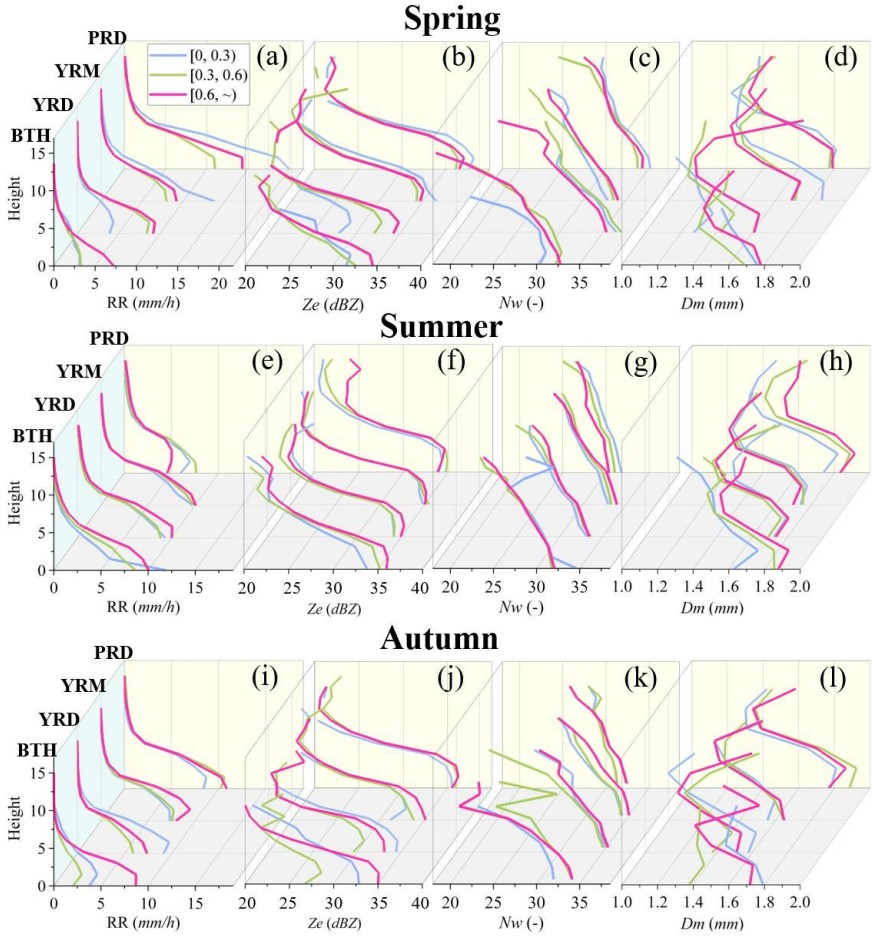

**Fig .4.** Vertical profiles of the average convective precipitation parameters $Z_e$, $D_m$, $N_w$

and RR for different regions in different seasons under three AOD conditions

( [0,0.3]–blue lines; [0.3,0.6]–green lines; [0.6, ~)–magenta lines ). Within each

subplot, line profiles are categorized into four regions: BTH, YRD, YRM, and PRD.

To enhance the characterization of the vertical structure and microphysical

processes of stratiform precipitation, Fig. S5 displays the mean vertical profiles of $Z_e$,

$D_m$, $N_w$, and RR, similar to those in Fig. 4. A notable $Z_e$ peak at an altitude of 5 km



corresponds to the 0 °C bright band signature, which is a distinctive indicator of the
hydrometeor phase transition and stratiform precipitation. Fig. S5a illustrates that the
PRD exhibits declining trends in RR and $Z_e$, diverging from the characteristics of
convective precipitation depicted in Fig. 4a–b. In the PRD, increasing aerosol loading
induces a Twomey effect: $N_w$ increases, accompanied by a decrease in $D_m$ (Fig.
S5c–d). This dominance of particle competition mechanisms at higher AOD loadings
aligns with increasing OCA contributions (Fig. S2c–d), as light–absorbing OCAs
suppress precipitation via semi–direct effects of shortwave radiation absorption.
During summer (Fig. S5 e–h), the BTH region displayed distinct patterns
compared to the other three regions: RR decreases with increasing aerosol loading,
whereas $Z_e$ and $D_m$ increase simultaneously. This suggests that higher aerosol loads
enhance particle albedo, thereby intensifying the evaporation of smaller particles and
the processes of break–up (Fig. 5b), such that while $N_w$ remains stable, the PEI
declines (Fig. S4j). Concurrently, the increased abundance of hygroscopic $SO_4A$
further depletes the atmospheric moisture. These combined effects lead to a notable
reduction in precipitation within the BTH region. The summer stratiform precipitation
responses of the PRD to aerosol loading resemble those of convective precipitation,
whereas the YRD and YRM show negligible alterations in the vertical profiles,
suggesting a low sensitivity of stratiform precipitation to aerosol loading.
**4 Influence of aerosols on precipitation microphysical processes**
To validate the aforementioned microphysical processes, this study assesses
near-surface precipitation mechanisms below the melting layer. The melting layer



refers to the region where the ice phase of the hydrometeors transitions to the liquid
phase during precipitation(Hu et al., 2024), and the analysis employs the
categorization approach established by Kumjian and Prat (2014). This approach
employs radar reflectivity ($\Delta Z_e = Z_e^{1km} - Z_e^{3km}$) and raindrop size ($\Delta D_m = D_m^{1km} - D_m^{3km}$)
differences between 1 km and 3 km above ground level. These metrics classify
processes into four categories: size sorting evaporation, coalescence, break-up, and
break-up coalescence balance. The extensive application of this methodology
demonstrates that coalescence and break-up processes dominate cloud microphysics
(Chen et al., 2025; Hu et al., 2022; Wen et al., 2023; Zhou et al., 2022).
Consequently, this analysis focuses exclusively on these two mechanisms. Fig. 5
displays the coalescence and break-up processes for convective and stratiform
precipitation across regions and seasons.

In convective precipitation (Fig. 5a), coalescence consistently dominates over

break-up across all regions, particularly during spring and autumn. With rising AOD
concentrations, the YRD, YRM, and PRD exhibit enhanced coalescence in spring,
whereas the PRD shows a decreasing-then-increasing trend in coalescence. In summer,
the proportions of coalescence and break-up remain comparable, whereas autumn
exhibits nonlinear responses in the BTH and YRD.

In stratiform precipitation (Fig. 5b), break-up generally exceeds coalescence,

with distinct seasonal patterns: in spring, the BTH exhibits increasing-then-decreasing
coalescence, whereas the PRD shows the opposite trend (aligning with divergent RR
patterns in Fig. S5a). During summer and autumn, BTH consistently shows notably



lower break-up than coalescence, which is one of the reasons why precipitation
continues to decline with increasing AOD loading in summer. The specific
microphysical influence process is explained in detail in Section 3.

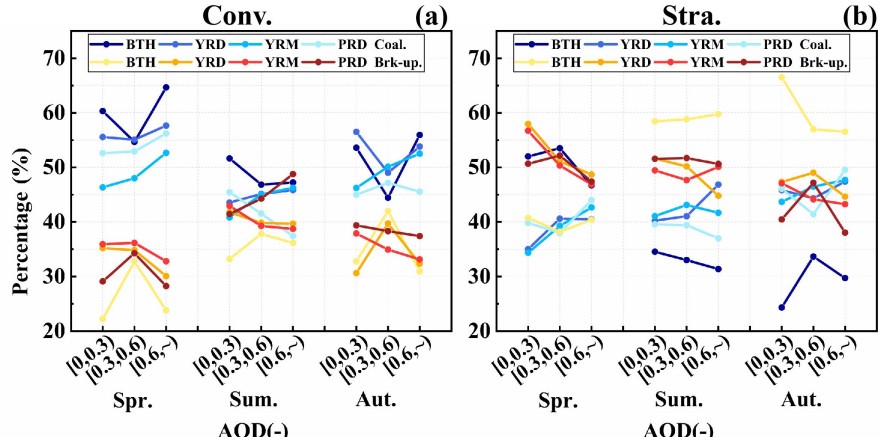


**Fig. 5.** Average point line plots of coalescence and break-up processes in precipitation

across different regions and seasons (Spr.-Spring, Sum.-Summer, and Aut.-Autumn)

under three AOD conditions. Here, (a) represents convective precipitation, and (b)

represents stratiform precipitation.

**5 Meteorological effects**
*5.1 Sensitivity analysis of aerosols to meteorological factors in precipitation*
*parameters*

Since precipitation processes are equally influenced by thermodynamic and

dynamic environments, this study employs RH at 850hPa as a thermal influence
factor and CAPE as a dynamic influence factor to examine aerosol sensitivity to these
meteorological elements. Following the classification criteria established in Section



2.5, RH and CAPE values were categorized into low, medium, and high levels.
Following the format of Fig. 3, Fig. 6 displayed point-line plots of convective
precipitation parameters across regions under different RH and aerosol loading
conditions. Notably, higher RH values consistently enhance mean precipitation
parameters (nsRR, LWP, PEI) across all regions, whereas STH and IWP show
inconsistent seasonal and regional variations. This suggests that elevated RH supplies
additional moisture, thereby mitigating moisture competition effects.
In spring (Fig. 6a–e), rising RH values do not interfere with the established
trends of the BTH across varying aerosol levels: nsRR and LWP continue to rise with
aerosol loading, yet STH and IWP exhibit persistent decrease-then-increase
trajectories. Precipitation parameters in the YRD and YRM remain consistent with the
characteristics observed in Fig. 3a–e. By contrast, the PRD displays distinct
characteristics under moderate RH conditions. This indicates that aerosols exhibit
greater sensitivity to RH in the spring convective precipitation of the PRD, whereas
aerosol effects dominate over RH influences in the BTH, YRD, and YRM.
Summer patterns (Fig. 6f–j) show modified the BTH responses under high RH;
however, other regions remain consistent with prior trends in Fig. 3f–j. In autumn,
observations (Fig. 6k–o) highlight inconsistent aerosol–precipitation relationships
across the RH levels in the PRD. By contrast, the BTH, YRD, and YRM maintain
nearly identical parameter responses.
The results indicate that RH sensitivity significantly influences
aerosol-precipitation effects on convective precipitation in the PRD, whereas aerosol



loading is the main factor affecting precipitation parameters in the other three regions.

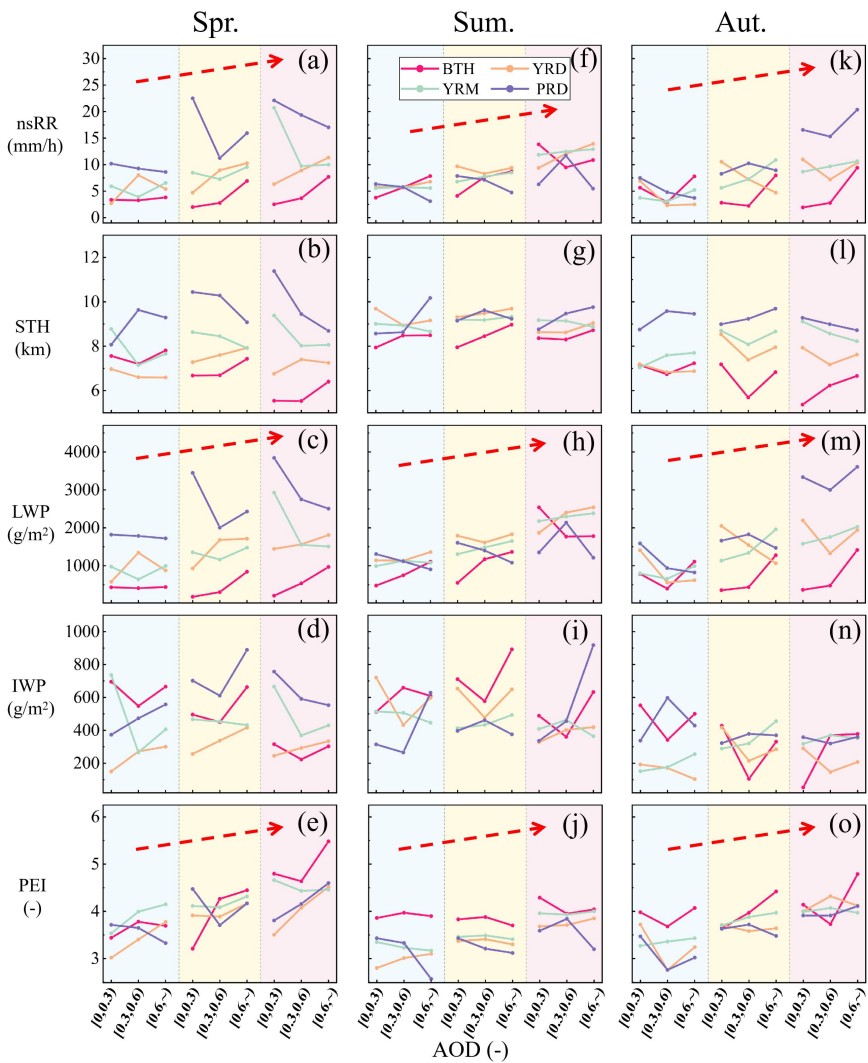


**Fig. 6.** Point-line graphs of mean values for convective precipitation parameters
(nsRR, STH, LWP, IWP, and PEI) across seasons and regions. The analysis is based
on three AOD intervals and various RH conditions. Each subpanel displays RH
gradients from left to right: low RH (blue background), medium RH (yellow
background), and high RH (red background). The red dashed arrow represents the





overall variation trend of precipitation parameters with the increase of RH.
Fig. S6 presents point-line plots illustrating mean precipitation parameters associated
with stratiform precipitation. Increased RH consistently enhances these parameters,
particularly nsRR, LWP, and PEI, in BTH, YRD, and YRM. However, the PRD
exhibits differing parameter responses across seasons under varying relative humidity
conditions, similar to convective precipitation, as aerosol loading rises.

In addition to the thermodynamic conditions, CAPE was selected as a dynamic

factor. Similar to the RH, the precipitation parameter characteristics across regions
were investigated under varying CAPE conditions. As indicated by the red dashed
arrows in Fig. 7, increasing CAPE values provide favorable dynamic conditions for
convective precipitation, leading to rising STH, increased IWP, and enhanced
ice-phase processes.

Spring (Fig. 7a–e) shows that in the PRD, the characteristics of nsRR, LWP, and

PEI under varying AOD loading remain consistent across different CAPE conditions.
In contrast, the BTH, YRD, and YRM exhibit distinct variations in these parameters
under different CAPE levels. Summer (Fig. 7f–j) reveals that aerosol effects in the
BTH region demonstrate heightened sensitivity to CAPE variations. However, during
autumn convective precipitation (Fig. 7k–o), aerosols across the four regions exhibit
substantial sensitivity to CAPE, indicating different seasonal response mechanisms to
atmospheric instability in these areas.

Fig. S7 presents point-line plots of the mean precipitation parameters for

stratiform precipitation. Increasing CAPE enhances the STH and IWP parameters,




which is consistent with the convective precipitation patterns in Fig. 6.

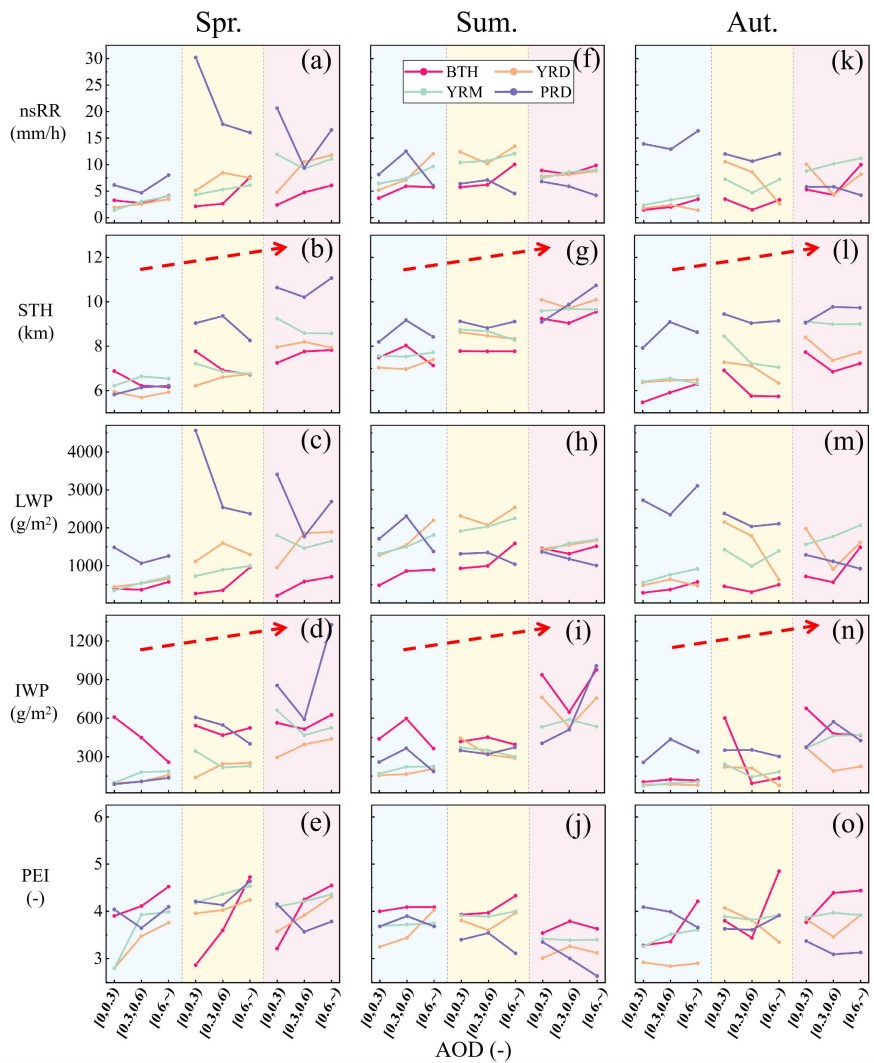


**Fig. 7.** Point-line graphs of mean values for convective precipitation parameters
across seasons and regions. The analysis is based on three AOD intervals and varying
CAPE scenarios. The form of this expression is similar to that shown in Fig. 6.
***5.2 Sensitivity analysis of aerosols to meteorological factors in the vertical structure***
***of precipitation***



Furthermore, to examine aerosol sensitivity to thermodynamic conditions and their effects on the vertical profiles of precipitation components, mean vertical profiles of precipitation parameters ($Z_e$, $D_m$, $N_w$, RR) at varying RH levels are illustrated, according to the technique shown in Fig. 4. Notably, as the parameter profile variations in Fig. 5 are concentrated within 0–10 km, the vertical coordinate range is limited to 0–10 km to emphasize the core precipitation processes. Moreover, where the curves intersect, dashed lines are employed to distinguish the selected profiles while maintaining the same representational integrity as the solid lines.

Spring convective precipitation (Fig. 8a–l) exhibits region-specific responses to RH, and in the BTH and YRD, increasing RH from low to medium ranges significantly elevates precipitation under high aerosol loading but suppresses it under medium loading (Fig. 8a). This discrepancy arises because abundant particles under elevated RH conditions undergo accelerated condensational growth, which increases $D_m$ (Fig. 8f). In contrast, the YRM and PRD show that RH enhancement primarily boosts precipitation, under low aerosol loading (blue curves). This is because in the YRD and YRM, particle competition continues to dominate at high loading, whereas added moisture at low loading facilitates condensational growth and enhances $N_w$ (Fig. 8g).

For the stratiform precipitation (Fig. 8m–x), the BTH, YRD, and YRM show consistent rightward shifts in RR curves across aerosol gradients as RH increases. This suggests that RH enhances moisture availability without modifying microphysical competition mechanisms. The PRD exhibits a response analogous to its



convective precipitation feature.

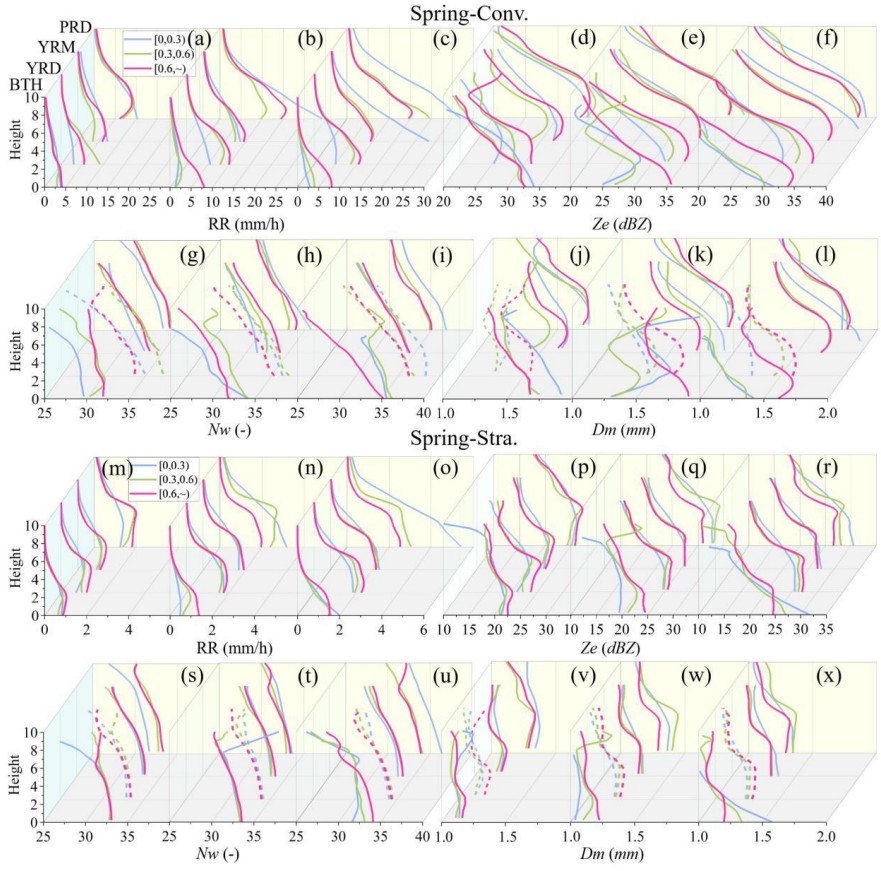

**Fig. 8.** Vertical profiles of mean precipitation parameters for convective (a–l) and
stratiform (m–x) precipitation in spring across four regions. The profiles are shown
under three AOD scenarios and RH conditions (arranged left to right as low, medium,
high RH; e.g., panels a–c correspond to low, medium, high RH, respectively). To
differentiate overlapping curves, selected profiles are plotted as dashed lines while
retaining the same representational validity as solid lines.

Additionally, similar characteristics are observed in convective and stratiform

precipitation during summer (Figs. S8) and autumn (Figs. S9) with variations in RH.



In general, increasing RH provides moisture conditions, accelerates cloud particle
condensational growth, and simultaneously increases both $D_m$ and $N_w$, thereby
enhancing precipitation. However, this process also depends on the content of CCN
and various physical competition mechanisms.

Similarly, vertical profiles of precipitation parameters under varying CAPE

conditions are presented. Fig. S10 illustrates consistent patterns between convective
and stratiform precipitation during spring, echoing the fundamental characteristics in
Fig. 8. This consistency suggests that RH and CAPE exert analogous influences on
precipitation across aerosol loading gradients during spring.

Summer convective precipitation (Fig. S11a–l) reveals distinct regional

responses. In the BTH region, CAPE elevation significantly enhances low-AOD
precipitation, likely driven by improved dynamic forcing that promotes cloud
development (red point line in Fig. S7g). In contrast, the PRD exhibits pronounced
precipitation suppression, most evident under moderate aerosol loading, where
heightened CAPE intensifies particle break-up processes (Fig. S13h). These findings
indicate that RH and CAPE exert divergent influences across regions. For RH,
increasing moisture availability promotes particle growth via condensation under
suited particle concentrations, but the Twomey effect dominates under high AOD
loading, where particle competition for cloud water prevails. CAPE provides
favorable dynamic conditions for cloud development, but simultaneously intensifies
particle break-up through dynamic forces, which hinders the constant growth of cloud
droplets and suppresses precipitation.



### 5.3 Sensitivity analysis of aerosols to meteorological factors in precipitation microphysical processes

To validate the aforementioned inferences, the proportions of break-up and coalescence processes in convective and stratiform precipitation are further investigated. Fig. S13 reveals that in convective precipitation, an increase in RH generally correlates with enhanced coalescence (white-green bars in the upper half; the trend is shown by the blue arrows) and reduced break-up (white-green bars in the lower half). Conversely, increasing CAPE is associated with decreased coalescence (green line in the upper half) and intensified break-up (yellow line in the lower half; the trend is shown by the red arrows), particularly in summer and the PRD region. As illustrated in Fig. S14, stratiform precipitation demonstrates similarities to convective precipitation, and the increase in RH makes the enhancement of coalescence processes more universal.




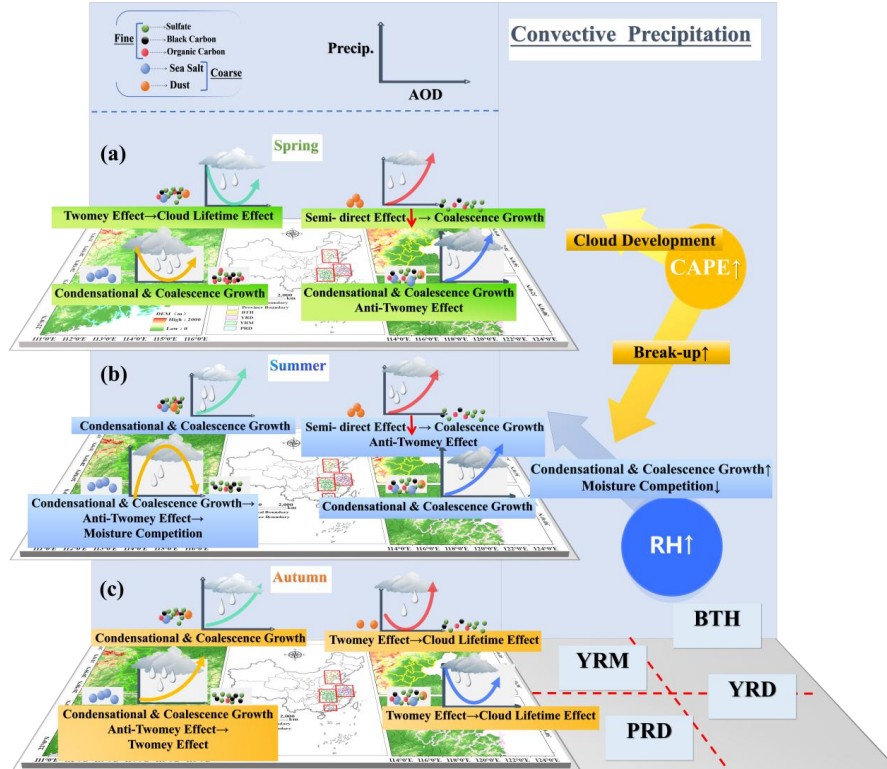

**Fig. 9.** Theoretical framework of aerosol impact on convective precipitation in the
BTH, YRD, YRM, and PRD: (a) spring, (b) summer, and (c) autumn. Symbol
conventions, ↑: Enhancement of process; ↓: Weakening of process; →: Transition
from left-side process dominance to right-side process dominance; Right-side CAPE
arrows: ↗ promotes precipitation; ↘ suppresses precipitation; Right-side RH arrows:
↗ enhances precipitation processes. Arrow length reflects the relative process
intensity.



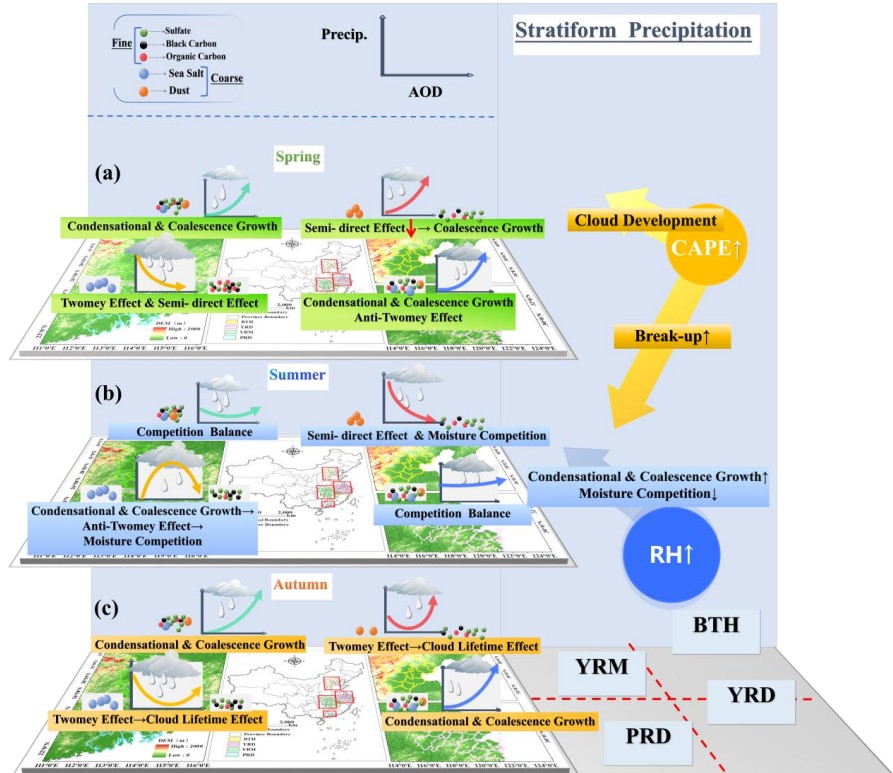

**Fig. 10.** Theoretical framework of aerosol impact on stratiform precipitation in the BTH, YRD, YRM, and PRD. The form of this expression is similar to that shown in Fig. 9.

**6 Conclusion**

This study systematically examined the impact of aerosols on precipitation parameters, vertical structures, and microphysical processes in convective and stratiform precipitation across China's four major urban clusters (the BTH, YRD, YRM, and PRD) — during spring, summer, and autumn, utilizing the DPR-MERRA-2-ERA5 dataset. It further explores aerosol sensitivity to RH and CAPE, revealing regional heterogeneity, seasonal dependency, and the underlying



microphysical processes of aerosol effects. The research indicates that physical
processes, including condensational growth, coalescence growth, semi-direct effects,
and moisture competition effects from aerosol-sourced CCN, trigger the Twomey
effect, Anti-Twomey effect, and cloud lifetime effect, resulting in varied precipitation
alterations. Additionally, an increase in aerosol loading diminishes the regional
disparities in precipitation characteristics, with a more pronounced effect during the
spring and summer. The precipitation parameters exhibit greater regional variability
than seasonal variability, and convective precipitation experiences more significant
seasonal and regional changes compared to stratiform precipitation. Based on the
findings in Section 3-5, the physical mechanisms by which aerosols at varying
concentrations influence convective precipitation (Fig. 9) and stratiform precipitation
(Fig. 10) are illustrated, with the following specific conclusions:
For convective precipitation (Fig. 9): Precipitation in the BTH region is
influenced by seasonal variations in dust aerosols. During spring (Fig. 9a) and
summer (Fig. 9b), dust aerosols exert significant impacts, whereas their contributions
declines in autumn (Fig. 9c), resulting in distinct precipitation characteristics.
Specifically, as the total aerosol concentration increases, the proportion of dust
aerosols rapidly decrease. This reduction weakens the semi-direct effect of dust while
enhancing the particle coalescence processes, thereby diminishing precipitation
suppression. However, insufficient moisture supply and frequent dust events in spring
collectively reduce the overall precipitation below the levels observed in the other
three regions. In autumn, when the DUA constitutes a minor fraction, rising aerosol



concentrations initially suppress precipitation through the Twomey effect, while
simultaneously promoting cloud development, subsequently enhancing precipitation
through increased CCN availability. The YRD exhibits a persistent precipitation
increase with increasing aerosol concentrations owing to the ample moisture supply.
While sharing similar seasonal trends with the BTH, its underlying mechanisms differ
significantly: abundant water vapor enables continuous precipitation growth during
spring (Fig. 9a) and summer (Fig. 9b), primarily attributable to enhanced droplet
condensation and coalescence processes. The PRD exhibits the most pronounced
seasonal variability, attributable to shifts in the composition of hygroscopic aerosols
(SSA). During spring (Fig. 9a), precipitation in the PRD is significantly higher than in
other regions under low aerosol loading due to SSAs. As aerosol concentrations
increase, diminishing SSA proportion weakens this enhancement until rising
hygroscopic organic carbon subsequently reinforces precipitation. In summer (Fig.
9b), sufficient moisture initially promotes droplet growth through
condensation-coalescence under low aerosol levels. However, the subsequent aerosol
accumulation intensifies moisture competition and suppresses precipitation.
Monsoon-influenced sea-salt overabundance (Xiao et al., 2025) further amplifies this
competition effect, resulting in overall lower precipitation rates compared to other
regions.
For stratiform precipitation (Fig. 10): Overall, stratiform and convective
precipitation share fundamental similarities yet exhibit distinct microphysical
processes due to differing cloud formation conditions. With a lower moisture supply



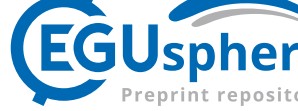

than convective systems, stratiform precipitation in the BTH region is suppressed
during summer (Fig. 10b) through aerosol semi-direct effects and moisture
competition. Similarly, in the PRD, spring precipitation is reduced by organic carbon
aerosols (Fig. 10a), which act as both hygroscopic and light-absorbing particles
(Zhuang et al., 2025). This occurs when an insufficient moisture supply enhances the
radiation-absorbing effect, dominating the precipitation reduction mechanism.
Furthermore, variations in RH and CAPE modulate aerosol-precipitation
interactions, as shown in Figs. 9–10. Specifically, elevated RH indicates enhanced
moisture availability, which facilitates rapid droplet growth through condensation and
coalescence under suitable aerosol loading. Regarding dynamic influences, increased
CAPE provides favorable conditions for cloud development while simultaneously
enhancing droplet break-up through intensified turbulence, hindering cloud droplet
growth, and suppressing precipitation, particularly in summer and the PRD region.
Overall, aerosol impacts on precipitation result from complex couplings among
regional aerosol composition, moisture transport patterns, atmospheric stability, and
precipitation types, generating both linear and nonlinear responses. These complex
dynamics establish essential theoretical underpinnings for formulating atmospheric
cleanup techniques in significant metropolitan centers, enhancing early warning
systems for extreme precipitation occurrences, and refining regional climate models.







**7 Discussion**
Building on the findings of Peng et al. (2025), which investigated the effects of
fine and coarse aerosols on summer precipitation structure and microphysics in the
YRD region, the present study expands the scope of analysis to examine aerosol
impacts on precipitation vertical structures and microphysical processes across
multiple regions and seasons in China. This extended scope has led to the following
new findings: (1)Enhanced aerosol loading reduces regional precipitation disparities,
most pronounced in spring and summer. (2)Precipitation exhibits stronger regional
than seasonal variability. (3)The BTH precipitation is dominated by dust aerosols,
whereas the YRD and PRD are influenced by sea salt aerosols. These conclusions are
primarily derived from analyses of satellite-based datasets, which provide extensive
spatial coverage, high spatiotemporal resolution, and continuous temporal monitoring.
Nevertheless, it is important to acknowledge that considerable uncertainties persist in
satellite data processing and retrieval algorithms, especially under complex
atmospheric and surface conditions. Additionally, spatiotemporal resolution and
format discrepancies across multisource data introduce unavoidable uncertainties.
This study primarily focuses on the vertical structural characteristics of precipitation,
whereas the analysis of aerosol data lacks comprehensive three-dimensional matching.
Currently, vertical profiling of aerosols relies primarily on aircraft sounding (Zhou et
al., 2023) and simulated radar signals (Fajardo-Zambrano et al., 2022), which remain
spatially limited. Satellite remote sensing is hindered by inadequate resolution and
deficiency in three-dimensional information (Li et al., 2022). However, the successful



launch and stable operation of EarthCARE now facilitates accurate three-dimensional
vertical profiling of clouds and aerosols via lidar (ATLID) and cloud profiling radar
(CPR) (Irbah et al., 2023). Future collaborative observations from the GPM and
EarthCARE will produce enhanced datasets on clouds, precipitation, and aerosols,
thus facilitating more robust in-depth studies within this research framework.
Subsequent research should integrate supplementary meteorological variables and
machine-learning methodologies to more effectively delineate aerosol effects and
examine their responsiveness to meteorological influences. Notably, as Zhao et
al.( 2025) revealed distinct aerosol-cloud interaction patterns over land versus ocean
in the YRD, the absence of cloud parameter products in this study may inherently
limit the depth of the aerosol-precipitation mechanism analysis. This methodological
constraint thus necessitates the future integration of high-resolution cloud parameter
datasets to refine research findings, enabling a comprehensive exploration of
aerosol-cloud-precipitation coupling mechanisms, specifically encompassing dry and
wet aerosol removal processes and precipitation feedback loops.
**Data availability**
The V07A GPM 2ADPR products used in this paper are openly available at the
NASA Goddard Space Flight Center's Precipitation Processing System (PPS) team
(https://storm.pps.eosdis.nasa.gov/ storm/).
**Author contributions**
H P & Z L: Writing – review & editing, Writing – original draft, Visualization,
Validation, Methodology, Investigation. X H: Writing – original draft, Validation,



Supervision, Software, Resources, Methodology, Investigation, Formal analysis, Data
curation, Conceptualization. W A: Writing – review & editing, Project administration.
S H & J Q: Writing – review & editing, Investigation. X Z: Writing – review &
editing, Funding acquisition, Formal analysis.
**Declaration of competing interest**
The authors declare that they have no known competing financial interests or
personal relationships that could have appeared to influence the work reported in this
paper.
**Acknowledgments**
Zhen Li and Heyuan Peng contributed equally to this work and should be
considered as co-first authors. The authors thank the anonymous reviewers for their
constructive comments and suggestions, which have greatly improved the quality of
this paper.
**Financial support**
This work has been jointly supported by the National Natural Science
Foundation of China (grant nos. 42305150).

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
