# Peer review of "Precipitation Microphysics and Vertical Structures over China's Megacity 2 3 Clusters Heyuan Peng1, Xiong Hu2\*, Weihua Ai1\*, Zhen Li1, Shensen Hu1, Junqi Qiao1, 4 5 Xianbin Zhao1 6 1College of Meteorology and O"

_EGUsphere, 2025_

## Referee Comment (RC1)

Comment to "Aerosol-Driven Precipitation Modification: Spatiotemporal Heterogeneity in Precipitation Microphysics and Vertical Structures over China's Megacity Clusters" by Peng et al.

This study investigates aerosol effects on precipitation vertical structures and microphysical characteristics across four Chinese urban clusters, including sensitivities to meteorological factors. Interesting results have been shown with suggested mechanisms. In principle, this study is worthy for publication with necessary modifications.

**Major comments:**

- (1) Noting the complex interactions between aerosol, cloud, precipitation, and meteorology, many physical explanations or mechanisms proposed here are with uncertainties, so cautions should be paid to the descriptions, particularly those claims regarding mechanisms.
- (2) As mentioned in the previous comment, the discussion of mechanisms appears somewhat limited. The mechanisms discussed in the study, such as the Twomey effect, lifetime effect, and semi-direct effect, are primarily associated with warm clouds. However, the precipitation cases analyzed in this study include both cold-topped and warm-topped systems. Important processes related to mixed-phase clouds, such as the role of ice nuclei and the invigoration effect (Rosenfeld et al., 2008), receive relatively little attention. For example, while the authors emphasize the importance of the semi-direct effect of dust in the BTH region, dust is also a major source of ice nuclei, which could substantially influence cold-topped precipitation formation. This aspect warrants further consideration and discussion.
- (3) If possible, a proofreading service is suggested to improve the English writing of this paper.

**Detailed comments:**

Line 45-46: IPCC report could be referred for this claim.

Line 48-51: In addition to early study, recent studies regarding their roles in hydrological cycle, extreme weather event, and climate system could be also mentioned as supporting material, such as Zhao et al. (2018, 2020, doi: 10.1029/2018GL079427, doi: 10.1093/nsr/nwz184).

Line 56: what does "these" here denote?

Line 62-65: Note that this effect is for absorptive aerosols within clouds, not for that outside the clouds.

Line 69-71: Regarding the regional variations, recent study by Li et al. (2025, doi: 10.1029/2024JD042649) could be mentioned.

Line 83-84: I am not sure if this claim is fair or not since many recent studies are actually considering the season, precipitation type, along with meteorological effects.

Line 151: remove "the" of "the 85%"

Figure 1: How do the authors consider the elevation effect on ACI? In other words, do the authors consider the spatial variation in elevation height when studying ACI at each region?

Line 172-175: We should also note that hail also often occurs in spring or even summer in north China.

Line 183-188: How reliable is the aerosol classification?

Line 189-190: How many hours are the aerosols used prior to precipitation events?

Line 191-198: similarly, the spatial and temporal resolution for data used here should be provided.

Line 197: A space is needed between the number and the unit: "850 hPa" instead of "850hPa" (also in Line 481).

Line 204: The definition of "valid precipitation systems" requires further clarification. It is unclear whether the threshold of four contiguous precipitation pixels refers to near-surface precipitation or to the existence of a precipitation profile. Some profiles may show precipitation aloft but none reaching the surface.

Line 208: "SO4A" should be "SO4A".

Line 232: Variable X should be defined.

Line 355-358: Relevant references should be added to support the statements.

Line 360: Fig. 5a mentioned here does not show RR and Ze; please check if this is a typographical error (should likely be "Fig. 4a").

Lines 361-362: The statement "... Ze increase linearly with aerosol loading in the BTH ..." seems valid only for the lower layers (<5 km) in the BTH according to Fig. 4b.

Line363-364: "Fig. 4a-e" may be a typographical error and should likely be "Fig. 3a-e".

Line 365: "Fig. S1a" should be corrected to "Fig. S2a".

Line 364-367: It seems that the authors did not catch the finding from cited study, what they indicated is: This discrepancy may arise from the predominant influence of DUA on the spring AOD composition of the BTH region, whereby their radiative effect enhances the atmospheric instability and triggers earlier precipitation.

Lines 386–387: It might be helpful to clarify whether the term "particle" refers to aerosol particles or to cloud/raindrop particles, as this term appears in multiple places throughout the manuscript. A clear distinction may improve the overall clarity of the physical interpretation.

Figure 4: Unit for height should be given (also in Fig. 8).

Lines 403-407: The sentence could be simplified to avoid redundancy and improve clarity.

Line 410-413: The statement that prolonged cloud lifetime enhances precipitation should be supported with appropriate references and further discussion.

Lines 435–437: Relevant references should be added to support the statement.

Line 453-455: I doubt if this claim is accurate or not. For example, there are almost no coalescence and break-up processes for non-precipitating clouds.

Line 499: please rephrase the sentence to make it clear.

Section 5.2 and 5.3, please modify the section title to make it more reasonable.

---

## Referee Comment (RC2)

This manuscript focuses on aerosol-driven precipitation modification over four major megacity clusters in China (BTH, YRD, YRM, PRD), systematically analyzing the spatiotemporal heterogeneity of precipitation microphysics and vertical structures using integrated datasets such as GPM DPR, MERRA-2, and ERA5. The research topic is highly relevant to regional hydrological cycles and climate model refinement, as aerosol-precipitation interactions in densely populated, polluted urban agglomerations remain a key uncertainty in atmospheric science. Overall, this manuscript is well organized: it investigates precipitation structural parameters, microphysical processes (e.g., coalescence, break-up), and the regulatory role of meteorological factors (RH, CAPE), providing a comprehensive framework for cross-regional comparisons. The conclusions regarding "regional precipitation disparities exceeding seasonal variations" and "aerosols mitigating regional differences in spring/summer" offer valuable insights for improving urban climate models. However, the manuscript has notable limitations that need to be addressed, including insufficient quantification of aerosol type contributions, lack of analysis on joint meteorological factor interactions, and minor issues in details (e.g., figures). With appropriate modifications to strengthen mechanistic analysis and technical rigor, the manuscript will likely meet the publication standards.

**Major comments:**

- 1) Quantitative contribution of different types of aerosols to precipitation lacks clarity: The manuscript claims that BTH's convective precipitation is dominated by dust aerosols (semi-direct effect), YRD/PRD by hygroscopic sea salt, and YRM by fine particles. However, it fails to provide quantitative data on the composition of aerosol types (DU, SS, SO4, BC, OC from MERRA-2) across seasons and regions. For example: No temporal-spatial maps of aerosol type proportions (e.g., the proportion of dust column mass concentration in spring over BTH) are provided to support the "dust-dominated semi-direct effect" conclusion.
- 2) I strongly recommend the authors supplement (1) seasonal/regional distribution maps of MERRA-2 aerosol type proportions; (2) correlation analyses between individual aerosol types (e.g., DU in BTH, SS in PRD)

- and precipitation parameters (e.g., RR, Dm); (3) absorption aerosol optical depth (AAOD) data to distinguish the role of absorbing (BC, DU) vs. scattering (SO4, SS) aerosols in microphysical processes. This will clarify the mechanism of aerosol type-specific impacts.
- 3) Ambiguity in precipitation type classification: The manuscript excludes "shallow convection" from convective precipitation based on 2ADPR criteria but does not specify the 2ADPR threshold for shallow convection and report the proportion of shallow convection in total precipitation across regions/seasons. This will affect sample representativeness and undermine the robustness of the findings.
- 4) It is well acknowledged that favorable meteorological conditions are indispensable for the formation and evolution of precipitation, thereby inevitably making it elusive to disentangle the aerosol effect on precipitation. This manuscript analyzes RH (thermodynamic) and CAPE (dynamic) separately but ignores their synergistic or antagonistic effects on aerosol-precipitation interactions. The readers are more willing to see does aerosol-induced coalescence strengthen more than in single-factor conditions under the high RH (sufficient moisture) + high CAPE (strong updrafts) conditions. The authors may conduct a two-factor crossed analysis (e.g., 3 RH levels × 3 CAPE levels) to quantify aerosol impacts on precipitation parameters under different combined meteorological scenarios. This will reveal the regulatory mechanism of thermodynamic-dynamic synergy, improving the comprehensiveness of the conclusion.
- 5) In Results and discussion part: I recommend adding one or two paragraphs to focus on key findings and providing a comparative discussion on how these findings align with or diverge from previous studies on aerosol-precipitation interaction. This will help more effectively highlight the study's unique contributions to the community.

**Minor comments**

- Lines 71-73: Except for the external synoptic conditions that can modulate the ACI process, entrainment, and other in-cloud meteorological factors, particularly that surrounding clouds and challenging to be measured, can affect the aerosol effect on clouds and precipitation.
- 2) Line 75: the citations are not correctly placed. Actually, these references are used to support "Significant research in recent years has focused

- on aerosol-induced modifications of precipitation structures in key regions of China".
- 3) The argument "analysis of specific seasons or precipitation types is frequently limited" is not correct. To my knowledge, the following references have investigated the aerosol effect on precipitation types, such as <a href="https://doi.org/10.1073/pnas.1715386115">https://doi.org/10.1029/2019GL085442</a>
- 4) Aerosol loading" and "aerosol concentration" are used interchangeably. The authors may unify to one terminology. And "vertical structures" and "vertical profiles" (used for Ze, Dm) can be unified to "vertical profiles" for consistency.
- 5) The discussion mentions EarthCARE's potential for aerosol-cloud vertical profiling but does not specify how its data (ATLID lidar aerosol profiles, CPR cloud profiles) will address the current study's limitation of "inadequate 3D aerosol matching". I recommend adding 1–2 sentences on future research directions (e.g., combining EarthCARE's aerosol vertical distribution with GPM DPR's precipitation profiles to analyze aerosol-altitude-dependent impacts on cloud microphysics), enhancing the discussion's innovations.
- 6) Figure 1 caption can be rephrased as "Geographical distribution of four urban agglomerations over (a) the Yangtze River Middle Reaches, (b) Beijing–Tianjin–Hebei, (c) Pearl River Delta, and (d) Yangtze River Delta (d), which is superimposed with elevation."
- 7) Fig. 3's y-axis for PEI lacks clarity—specify that "-" indicates dimensionless (after scaling by 1000). Also, "across the FOUR regions and seasons." is not correct, I only saw three seasons corresponding to three rows of line plots.
- 8) Figure 5's title for X-axis lacks clarity—specify that "-" indicates dimensionless or directly delete "-" (as we all know AOD is dimensionless).
- 9) Figures 2-3, 5-7: Spr. Sum. and Aut are not standard abbreviation for three different seasons.
- 10) "Conclusion" -> "Conclusions"

---

## Author Comment (AC1)

**Responses to Reviewers' Comments**

We are sincerely grateful to the editor and reviewers for their valuable time for reviewing our manuscript. The comments are very helpful and valuable, and we have addressed the issues raised by the reviewer in the revised manuscript. Please find our point-by-point response (in blue text) to the comments (in black text) raised by the reviewer. We have revised the paper according to your comments (**highlighted in blue text of the revised manuscript**).

**Reviewer #1:**

This study investigates aerosol effects on precipitation vertical structures and microphysical characteristics across four Chinese urban clusters, including sensitivities to meteorological factors. Interesting results have been shown with suggested mechanisms. In principle, this study is worthy for publication with necessary modifications.

**Response:**

Thank you for your thorough review and constructive feedback on our manuscript titled *"Aerosol-Driven Precipitation Modification: Spatiotemporal Heterogeneity in Precipitation Microphysics and Vertical Profiles over China's Megacity Clusters"*. We appreciate the time and effort you have invested in understanding our work and for providing us with clear directions for improvement. We confirm that all your earlier concerns were addressed in detail in our initial point-by-point response letter (submitted with the revised manuscript) and implemented in both the tracked-changes and clean versions of the manuscript. Below are our point-by-point responses:

**Major comments:**

[1].Noting the complex interactions between aerosol, cloud, precipitation, and meteorology, many physical explanations or mechanisms proposed here are with uncertainties, so cautions should be paid to the descriptions, particularly those claims regarding mechanisms.

**Response:**

We fully agree with your insightful comment regarding the uncertainties in the proposed mechanisms. Aerosol-cloud-precipitation interactions are inherently complex, involving nonlinear couplings between multiple factors (e.g., aerosol type, vertical distribution, meteorological conditions). To address this concern comprehensively, we have not only refined the manuscript's language and supplemented literature but also conducted additional quantitative analyses to strengthen the robustness of our mechanistic inferences. Specifically:

(1) Language refinement: we adopted more cautious phrasing to avoid overstated causal claims. For instance, we replaced absolute terms such as **"dominate"** with **"significantly influence"** or **"key"**.

(2) Evidence supplementation: We integrated new analyses to quantify aerosol-type-specific impacts, including:

(i) seasonal/regional distribution maps of MERRA-2 aerosol type proportions **(Figs. S3–S8 in the revised SI)**, which visually confirm the dominance of dust aerosols (DUA) in the BTH and sea salt aerosols (SSA) in the PRD;

These analyses provide quantitative support for our mechanistic claims, reducing reliance on qualitative inference. The corresponding figure citations and discussions for these findings have been incorporated into the main text of the revised manuscript. For instance:

**"The spatial and seasonal distributions of aerosol composition over China's megacity clusters, are detailed in Figs. S3-S8. Overall, the BTH region is characterized by a high burden of DUA (Figs. S3-S8b), particularly during spring. In contrast, the PRD region shows a pronounced signal of SSA, especially in spring and summer (Figs. S3-S8d). Figs. S3-8 (f-g) show that loadings of both absorbing and scattering aerosols are markedly elevated in the BTH region, in contrast to the discernibly lower concentrations observed in the PRD. These characteristic distributions provide a fundamental basis for interpreting the region-specific aerosol-precipitation mechanisms discussed in subsequent sections" (Section 2.3, Lines 198-206)**

**"In the BTH region, which is characterized by a high concentration of DUA (Figs. S2-8). Sun and Zhao (2021) attributed similar phenomena to the radiative effect of aerosols enhancing atmospheric instability and triggering earlier precipitation." (Section 3.3, Lines 406-409)**

[Figure]

**Fig. S3.** Spatial distribution map of seven types (BC, DU, OC, SS, SO$_4$, absoring, and scattering) of average aerosols in spring from 2014 to 2023 over China's Megacity Clusters ( BTH, YRD, YRM, and PRD).

[Figure]

**Fig. S4.** Spatial distribution map of seven types (BC, DU, OC, SS, SO₄, absoring, and scattering) of aerosol type proportions in spring from 2014 to 2023 over China's Megacity Clusters (BTH, YRD, YRM, and PRD).

[Figure]

**Fig. S5.** Spatial distribution map of seven types (BC, DU, OC, SS, SO₄, absoring, and scattering) of average aerosols in summer from 2014 to 2023 over China's Megacity Clusters ( BTH, YRD, YRM, and PRD).

[Figure]

**Fig. S6.** Spatial distribution map of seven types (BC, DU, OC, SS, SO₄, absoring, and scattering) of aerosol type proportions in summer from 2014 to 2023 over China's Megacity Clusters (BTH, YRD, YRM, and PRD).

[Figure]

**Fig. S7.** Spatial distribution map of seven types (BC, DU, OC, SS, SO₄, absoring, and scattering) of average aerosols in autumn from 2014 to 2023 over China's Megacity Clusters ( BTH, YRD, YRM, and PRD).

[Figure]

**Fig. S8.** Spatial distribution map of seven types (BC, DU, OC, SS, SO₄, absoring, and scattering) of aerosol type proportions in autumn from 2014 to 2023 over China's Megacity Clusters (BTH, YRD, YRM, and PRD).

(ii) Spearman correlation analyses between key aerosol types (DUA in BTH, SSA in PRD) and precipitation parameters (RR, $N_w$, $D_m$; **Fig. S12 in the revised SI**), which statistically validate the link between specific aerosols and microphysical processes;

The newly added correlation analyses figure (Fig. S12) further elucidate the distinct seasonal and precipitation type dependent roles of key aerosol species. We have added the sentence to specify that: **"Despite a complex influence that varies with season and precipitation type, DUA in the BTH region exhibit a consistent negative correlation with $N_w$ (Fig. S12b-e). This negative correlation could be attributed to the semi-direct effect of DUA, whereby the**

absorption of solar radiation heats the atmosphere, thus promoting cloud droplet evaporation and suppressing droplet concentrations. The negative relationship with RR and $D_m$ is particularly evident in stratiform precipitation under low-to-medium aerosol loadings (Fig. S12d). For convective precipitation, the influence of DUA is seasonally modulated, showing a negative correlation in summer but a positive correlation in spring and autumn. Meanwhile, in the PRD region (Fig. S12g-l), SSA during spring and summer are consistently positively correlated with precipitation parameters. These statistically significant relationships further corroborate the proposed type-specific mechanisms." (Section 3.3. Lines 391-402)

[Figure]

**Fig. S12.** Spearman correlation coefficients between key aerosol types (DUA for the BTH and SSA for the PRD) and precipitation parameters (RR, $N_w$ and $D_m$) across regions and seasons under the three AOD regimes. Color gradients (from yellow to blue) encode the correlation strength and direction, and asterisks denote statistical significance (*: $p<0.05$, **: $p<0.01$, ***: $p<0.001$)..

(iii) Differentiation of absorbing (BCA, DUA) vs. scattering (SO$_4$A, SSA) aerosols, which clarifies how aerosol radiative properties modulate cloud-precipitation interactions. (**Figs. S3–S8 in the revised SI**)

(3) Literature and uncertainty discussion: We supplemented relevant studies (e.g., Rosenfeld et al., 2008; Li et al., 2025; Zhao et al.,2025) to contextualize our findings within existing knowledge and explicitly discussed uncertainties in the Discussion section, including limitations in 3D aerosol-cloud co-location and potential synergies between meteorological factors. We also highlighted future directions (e.g., integrating EarthCARE's ATLID/CPR data with GPM DPR) to address these gaps.

We have revised and added sentences to specify that **(7 Discussion)**:

**"The results provide further evidence for several established mechanisms: The dominant role of dust aerosols in the BTH region aligns with existing research (Xi et al., 2024; Xiao et al.,2025), manifesting as impacts on precipitation through ARI (Sun and Zhao, 2022) and semi-direct effects, and also serving as effective IN to promote convective cloud development (Xi et al.,2025). Furthermore, the precipitation-enhancing effect of sea salt aerosols in the PRD region is consistent with prior observations (Guo et al., 2022; Chen et al.,2025). (Section 7. Lines 757-763)**

**"However, as noted by Stier et al. (2024) and Zhao et al. (2024), aerosol impacts on precipitation remain highly complex, and their net effects are still subject to considerable uncertainty across different scales. Multiple factors are known to modulate precipitation processes, such as vertical wind shear (Riemer et al., 2010), cloud properties (Shao and Liu, 2005; Zhao et al., 2012), and latent heating (Zhu et al., 2025). Therefore, a critical challenge for future research lies in better disentangling the influence of such environmental meteorological factors from the overall aerosol effect." (Section 7. Lines 764-771)**

By incorporating linguistic rigor, supplementary quantitative data, and a frank uncertainty analysis, we have ensured that our mechanistic interpretations are both evidence-based and appropriately constrained within the observational scope of this study.

We are grateful for this valuable suggestion again! Your comments have been instrumental in driving a significant upgrade to our manuscript. The new figures and correlation analyses provide the clear, quantitative evidence needed to solidify our conclusions regarding aerosol type-specific impacts.

**References:**

Chen, F., Yang, Y., Yu, L., Li, Y., Liu, W., Liu, Y., and Lolli, S.: Distinct effects of fine and coarse aerosols on microphysical processes of shallow-precipitation systems in summer over southern China, Atmos. Chem. Phys., 25, 1587–1601, https://doi.org/10.5194/acp-25-1587-2025, 2025.

Guo, J., Luo, Y., Yang, J., Furtado, K., and Lei, H.: Effects of anthropogenic and sea salt aerosols on a heavy rainfall event during the early-summer rainy season over coastal Southern China, Atmospheric Research, 265, 105923, https://doi.org/10.1016/j.atmosres.2021.105923, 2022.

Riemer, M., Montgomery, M. T., and Nicholls, M. E.: A new paradigm for intensity modification of tropical cyclones: Thermodynamic impact of vertical wind shear on the inflow layer, Atmos. Chem. Phys., 10, 3163–3188, https://doi.org/10.5194/acp-10-3163-2010, 2010.

Rosenfeld, D., Lohmann, U., Raga, G. B., O'Dowd, C. D., Kulmala, M., Fuzzi, S., Reissell, A., and Andreae, M. O.: Flood or Drought: How Do Aerosols Affect Precipitation?, Science, 321, 1309–1313, https://doi.org/10.1126/science.1160606, 2008.

Shao, H. and Liu, G.: Why is the satellite observed aerosol's indirect effect so variable?, Geophysical Research Letters, 32, https://doi.org/10.1029/2005GL023260, 2005.

Stier, P., van den Heever, S. C., Christensen, M. W., Gryspeerdt, E., Dagan, G., Saleeby, S. M., Bollasina, M., Donner, L., Emanuel, K., Ekman, A. M. L., Feingold, G., Field, P., Forster, P., Haywood, J., Kahn, R., Koren, I., Kummerow, C., L'Ecuyer, T., Lohmann, U., Ming, Y., Myhre, G., Quaas, J., Rosenfeld, D., Samset, B., Seifert, A., Stephens, G., and Tao, W.-K.: Multifaceted aerosol effects on precipitation, Nat. Geosci., 17, 719–732, https://doi.org/10.1038/s41561-024-01482-6, 2024.

Sun, Y. and Zhao, C.: Distinct impacts on precipitation by aerosol radiative effect over three different megacity regions of eastern China, Atmos. Chem. Phys., 21, 16555–16574, https://doi.org/10.5194/acp-21-16555-2021, 2021.

Xi, J., Li, R., Fan, X., and Wang, Y.: Aerosol effects on the three-dimensional structure of organized precipitation systems over Beijing-Tianjin-Hebei region in summer, Atmospheric Research, 298, 107146, https://doi.org/10.1016/j.atmosres.2023.107146, 2024.

Xi, J., Wang, Y., Li, R., Wu, B., Fan, X., Ma, X., and Meng, Z.: The impact of Sahara dust aerosols on the three-dimensional structure of precipitation systems of different sizes in spring, EGUsphere, 1–30, https://doi.org/10.5194/egusphere-2025-2799, 2025.

Xiao, Y., Zhang, J., Zhu, J., and Dai, Q.: Exploration of aerosol-precipitation relationships under different climate regimes in China, GIScience & Remote Sensing, 62, https://doi.org/10.1080/15481603.2025.2457992, 2025.

Zhao, C., Klein, S. A., Xie, S., Liu, X., Boyle, J. S., and Zhang, Y.: Aerosol first indirect effects on non-precipitating low-level liquid cloud properties as simulated by CAM5 at ARM sites, Geophysical Research Letters, 39, https://doi.org/10.1029/2012GL051213, 2012.

Zhao, C., Sun, Y., Yang, J., Li, J., Zhou, Y., Yang, Y., Fan, H., and Zhao, X.: Observational evidence and mechanisms of aerosol effects on precipitation, Science Bulletin, 69, 1569–1580, https://doi.org/10.1016/j.scib.2024.03.014, 2024.

Zhu, H., Zhao, H., Yang, S., Zhou, R., Wang, Y., Zou, Y., Zhao, C., and Li, R.: Smoke aerosols elevate precipitation top and latent heat to the upper atmosphere globally, npj Clim Atmos Sci, 8, https://doi.org/10.1038/s41612-025-01047-3, 2025.

[2].As mentioned in the previous comment, the discussion of mechanisms appears somewhat limited. The mechanisms discussed in the study, such as the Twomey effect, lifetime effect, and semi-direct effect, are primarily associated with warm clouds. However, the precipitation cases analyzed in this study include both cold-topped and warm-topped systems. Important processes

related to mixed-phase clouds, such as the role of ice nuclei and the invigoration effect (Rosenfeld et al., 2008), receive relatively little attention. For example, while the authors emphasize the importance of the semi-direct effect of dust in the BTH region, dust is also a major source of ice nuclei, which could substantially influence cold-topped precipitation formation. This aspect warrants further consideration and discussion.

**Response:**

We thank you for this insightful comment. We have expanded the discussion on mixed-phase cloud processes, particularly the role of ice nuclei (IN) and the invigoration effect, as follows:

(1) Added analysis on dust as IN: We conducted additional correlation analyses between DUA and ice-phase parameters (IWP, STH). The results confirm that dust aerosols enhance ice-phase processes in convective precipitation, supporting the invigoration effect (Rosenfeld et al.,2008). This analysis is now included in Section 3.1 and illustrated in new Fig. S10.

The results reveal a notable dual role of dust aerosols:

**"Given the dominant roles of DUA in BTH and SSA in PRD (Figs. S2-S8), this study further investigated how these key aerosol types distinctly modulate precipitation. Correlation analyses reveal that DUA is significantly positively correlated with both IWP and STH in the convective precipitation (Fig. S10), indicating an invigoration of ice-phase processes. This suggests that DUA, acting as efficient IN, promote the glaciation and vertical development of convective clouds, consistent with the invigoration effect (Rosenfeld., 2008). In contrast, the impact of DUA on stratiform precipitation is substantially weaker and more variable. This promotional mechanism, facilitated by ice-nucleating ability of DUA, stands in contrast to precipitation-suppressing semi-direct effect, highlighting the complex and multi-faceted nature of DUA impacts on different precipitation types." (Section 3.1. Lines 306-316)**

[Figure]

**Fig. S10.** Spearman correlation coefficients between DUA and precipitation parameters (STH, and IWP) in BTH for convective and stratiform precipitation under three AOD regimes and seasons.

(2) Revised mechanistic discussion: In 6 Conclusions, we now explicitly discuss the dual role of dust aerosols: (i) semi-direct effect (warming and suppression) and (ii) IN-mediated

invigoration (enhancement of ice-phase precipitation). This adds depth to our interpretation of regional differences, especially in BTH.

We have revised and added sentences to specify that **(6 Conclusions)**: **"Specifically, DUA play a dual role: it suppresses precipitation through the semi-direct effect (by evaporating cloud droplets), yet also invigorates deep convection by serving as efficient IN. This competitive dynamic between the suppression of warm-rain processes and the invigoration of cold-rain processes is central to the complex aerosol-precipitation relationship." (Section 6. Lines 696-700)**

We acknowledge that our original discussion did not fully capture the complexity of aerosol impacts across different cloud phases. Our analysis intentionally focused on the microphysics pivotal to the final precipitation output, particularly the coalescence and break-up processes below the melting layer, as these near-surface mechanisms directly govern the drop size distribution and intensity of rainfall reaching the ground. While the current study and the aforementioned additions have strengthened the analysis of ice-phase processes, a truly comprehensive dissection quantifying aerosol effects separately on the ice layer, melting layer, and liquid layer remains a challenge within the scope of this observational framework. In response, we have now outlined a clear pathway for future research in the 7 Discussion, emphasizing the need for advanced methodologies to disentangle aerosol impacts on precipitation microphysics across all hydrometeor phases.

**"The integration of high-precision vertical profiles from ATLID (aerosols), CPR (clouds), and GPM DPR (precipitation) will enable future researchers to quantify how aerosol layers at different altitudes modulate cloud microphysics and precipitation formation (Li et al.,2025b). For instance, this multi-source dataset provides an unprecedented opportunity to systematically unravel the distinct roles aerosols play in the microphysics of different hydrometeor phases, including the ice layer, melting layer, and liquid layer." (Section 7. Lines 784-790)**

We are grateful for this valuable suggestion again, which provides a clear and important direction for our subsequent researches.

Li, Z., Ge, S., Hu, X., Ai, W., Tang, J., Qiao, J., Hu, S., Zhao, X., Wu, H., Li, Z., Ge, S., Hu, X., Ai, W., Tang, J., Qiao, J., Hu, S., Zhao, X., and Wu, H.: Preliminary analysis of a novel spaceborne pseudo tripe-frequency radar observations on cloud and precipitation: EarthCARE CPR-GPM DPR coincidence dataset, Remote Sens., 17, https://doi.org/10.3390/rs17152550, 2025b.

Rosenfeld, D., Lohmann, U., Raga, G. B., O'Dowd, C. D., Kulmala, M., Fuzzi, S., Reissell, A., and Andreae, M. O.: Flood or Drought: How Do Aerosols Affect Precipitation?, Science, 321, 1309–1313, https://doi.org/10.1126/science.1160606, 2008.

[3].If possible, a proofreading service is suggested to improve the English writing of this paper.

**Response:**

We sincerely thank you for this important suggestion regarding the English writing. We fully acknowledge that the clarity and quality of the language are paramount for effective scientific communication. In direct response to this comment, we have undertaken a rigorous, multi-round revision process to thoroughly improve the manuscript.

This effort focused on key areas such as: (1) enhancing linguistic precision by ensuring consistent use of core terminology (e.g., **"aerosol loading,"** **"vertical profiles"**); (2) strengthening grammatical rigor by explicitly clarifying ambiguous pronoun references (including instances of "this" and "these" noted in the review); (3) simplifying complex sentence structures to improve overall readability and flow.

We are confident that this dedicated effort has significantly improved the manuscript's clarity, flow, and overall presentation. We are truly grateful for this constructive suggestion, which has undoubtedly strengthened the final quality of our work.

**Detailed comments:**

[1].Line 45-46: IPCC report could be referred for this claim.

**Response:**

Done. Thank you. We have added the citation to the (IPCC ,2013)**. "Aerosols modulate clouds and precipitation primarily through Aerosol–Radiation Interactions (ARI) and Aerosol–Cloud Interactions (ACI) (IPCC,2013)." (Lines 43-44)**

IPCC. 2013. Climate Change 2013: The Physical Science Basis. Contribution of Working Group I to the Fifth Assessment Report of the Intergovernmental Panel on Climate Change [M].Stocker TF,QinD, PlattnerG K, et al.,Eds.Cambridge, United Kingdom and New York, NY, USA: Cambridge UniversityPress, 1535pp.

[2].Line 48-51: In addition to early study, recent studies regarding their roles in hydrological cycle, extreme weather event, and climate system could be also mentioned as supporting material, such as Zhao et al. (2018, 2020, doi: 10.1029/2018GL079427, doi: 10.1093/nsr/nwz184).

**Response:**

Thank you for the suggestion. We have added the recommended references Zhao et al. (2018, 2020) to support this statement. **"These processes involve complex multiscale, multi-factor coupling effects with profound implications for regional hydrological cycles, extreme**

**weather events, and climate systems (Li et al., 2016, 2019; Ramanathan et al., 2001; Zhao et al., 2018, 2020)." (Lines 46-49)**

Li, Z., Lau, W. K.-M., Ramanathan, V., Wu, G., Ding, Y., Manoj, M. G., Liu, J., Qian, Y., Li, J., Zhou, T., Fan, J., Rosenfeld, D., Ming, Y., Wang, Y., Huang, J., Wang, B., Xu, X., Lee, S.-S., Cribb, M., Zhang, F., Yang, X., Zhao, C., Takemura, T., Wang, K., Xia, X., Yin, Y., Zhang, H., Guo, J., Zhai, P. M., Sugimoto, N., Babu, S. S., and Brasseur, G. P.: Aerosol and monsoon climate interactions over asia, Reviews of Geophysics, 54, 866 – 929, https://doi.org/10.1002/2015RG000500, 2016.

Li, Z., Wang, Y., Guo, J., Zhao, C., Cribb, M. C., Dong, X., Fan, J., Gong, D., Huang, J., Jiang, M., Jiang, Y., Lee, S.-S., Li, H., Li, J., Liu, J., Qian, Y., Rosenfeld, D., Shan, S., Sun, Y., Wang, H., Xin, J., Yan, X., Yang, X., Yang, X., Zhang, F., and Zheng, Y.: East asian study of tropospheric aerosols and their impact on regional clouds, precipitation, and climate (EAST-AIRCPC), Journal of Geophysical Research: Atmospheres, 124, 13026 – 13054, https://doi.org/10.1029/2019JD030758, 2019.

Ramanathan, V., Crutzen, P. J., Kiehl, J. T., and Rosenfeld, D.: Aerosols, Climate, and the Hydrological Cycle, Science, 294, 2119–2124, https://doi.org/10.1126/science.1064034, 2001.

Zhao, C., Lin, Y., Wu, F., Wang, Y., Li, Z., Rosenfeld, D., and Wang, Y.: Enlarging Rainfall Area of Tropical Cyclones by Atmospheric Aerosols, Geophysical Research Letters, 45, 8604–8611, https://doi.org/10.1029/2018GL079427, 2018.

Zhao, C., Yang, Y., Fan, H., Huang, J., Fu, Y., Zhang, X., Kang, S., Cong, Z., Letu, H., and Menenti, M.: Aerosol characteristics and impacts on weather and climate over the Tibetan Plateau, Natl Sci Rev, 7, 492–495, https://doi.org/10.1093/nsr/nwz184, 2020.

[3].Line 56: what does "these" here denote?

**Response:**

The word "these" referred to the "cloud microphysical processes". We have rephrased the sentence to **"Specifically, these modified cloud microphysical processes include cloud droplet spectrum distribution, phase transition efficiency, and precipitation formation pathways (Xie et al., 2013)" for better clarity. (Lines 55-57)**

Xie, X., Liu, X., Peng, Y., Wang, Y., Yue, Z., and Li, X.: Numerical simulation of clouds and precipitation depending on different relationships between aerosol and cloud droplet spectral dispersion, Tellus B: Chemical and Physical Meteorology, 65, 19054, https://doi.org/10.3402/tellusb.v65i0.19054, 2013.

[4].Line 62-65: Note that this effect is for absorptive aerosols within clouds, not for that outside the clouds.

**Response:**

We appreciate this clarification. We have revised the sentence to specify that: **"The Semi-Direct Effect shortens cloud lifetime through another mechanism: absorptive aerosols within clouds heat the atmosphere by absorbing shortwave radiation, thus promoting droplet evaporation". (Lines 62-65)**

[5].Line 69-71: Regarding the regional variations, recent study by Li et al. (2025, doi: 10.1029/2024JD042649) could be mentioned.

**Response:**

Done. We have cited the suggested reference Li et al. (2025). **"......leading to pronounced regional variations (Li et al.,2025a; Xiao et al., 2025)." (Line 71)**

Li, J., Zhao, C., Sun, Y., Zhao, X., Yang, J., Yang, Y., Chen, A., and Zhou, Y.: Distinct Aerosol Impacts on Local Scale Convective Rainfall Between Sichuan Basin and North China Plain Regions in China, Journal of Geophysical Research: Atmospheres, 130, e2024JD042649, https://doi.org/10.1029/2024JD042649, 2025a

Xiao, Y., Zhang, J., Zhu, J., and Dai, Q.: Exploration of aerosol-precipitation relationships under different climate regimes in China, GIScience & Remote Sensing, 62, https://doi.org/10.1080/15481603.2025.2457992, 2025.

[6].Line 83-84: I am not sure if this claim is fair or not since many recent studies are actually considering the season, precipitation type, along with meteorological effects.

**Response:**

We thank you for this comment. We have revised the statement to more accurately reflect the existing research landscape: "**Previous research has been conducted on aerosol-driven precipitation patterns in more general areas, such as the North China Plain (Sun et al., 2023), South China (Chen et al., 2025), and East China (Wen et al., 2023) under different seasonal and precipitation type conditions (Day et al., 2018; Guo et al., 2019). These studies have consistently confirmed that the modulation effects of aerosols on clouds and precipitation exhibit pronounced regional heterogeneity (Guo et al., 2017, 2019; Li et al., 2019; Sun and Zhao, 2021). However, the underlying physical mechanisms, particularly the cloud microphysical processes responsible for these disparate regional responses, are not yet fully understood.**" (Lines 82-90)

Chen, F., Yang, Y., Yu, L., Li, Y., Liu, W., Liu, Y., and Lolli, S.: Distinct effects of fine and coarse aerosols on microphysical processes of shallow-precipitation systems in summer over southern China, Atmos. Chem. Phys., 25, 1587 – 1601,

https://doi.org/10.5194/acp-25-1587-2025, 2025.

Day, J. A., Fung, I., and Liu, W.: Changing character of rainfall in eastern China, 1951–2007, Proceedings of the National Academy of Sciences, 115, 2016–2021, https://doi.org/10.1073/pnas.1715386115, 2018.

Guo, J., Su, T., Li, Z., Miao, Y., Li, J., Liu, H., Xu, H., Cribb, M., and Zhai, P.: Declining frequency of summertime local-scale precipitation over eastern China from 1970 to 2010 and its potential link to aerosols, Geophysical Research Letters, 44, 5700–5708, https://doi.org/10.1002/2017GL073533, 2017.

Guo, J., Su, T., Chen, D., Wang, J., Li, Z., Lv, Y., Guo, X., Liu, H., Cribb, M., and Zhai, P.: Declining Summertime Local-Scale Precipitation Frequency Over China and the United States, 1981–2012: The Disparate Roles of Aerosols, Geophysical Research Letters, 46, 13281–13289, https://doi.org/10.1029/2019GL085442, 2019.

Li, Z., Wang, Y., Guo, J., Zhao, C., Cribb, M. C., Dong, X., Fan, J., Gong, D., Huang, J., Jiang, M., Jiang, Y., Lee, S.-S., Li, H., Li, J., Liu, J., Qian, Y., Rosenfeld, D., Shan, S., Sun, Y., Wang, H., Xin, J., Yan, X., Yang, X., Yang, X., Zhang, F., and Zheng, Y.: East asian study of tropospheric aerosols and their impact on regional clouds, precipitation, and climate (EAST-AIRCPC), Journal of Geophysical Research: Atmospheres, 124, 13026–13054, https://doi.org/10.1029/2019JD030758, 2019.

Sun, Y., Wang, Y., Zhao, C., Zhou, Y., Yang, Y., Yang, X., Fan, H., Zhao, X., and Yang, J.: Vertical Dependency of Aerosol Impacts on Local Scale Convective Precipitation, Geophysical Research Letters, 50, https://doi.org/10.1029/2022gl102186, 2023.

Sun, Y. and Zhao, C.: Distinct impacts on precipitation by aerosol radiative effect over three different megacity regions of eastern China, Atmos. Chem. Phys., 21, 16555–16574, https://doi.org/10.5194/acp-21-16555-2021, 2021.

Wen, L., Chen, G., Yang, C., Zhang, H., and Fu, Z.: Seasonal variations in precipitation microphysics over East China based on GPM DPR observations, Atmospheric Research, 293, 106933, https://doi.org/10.1016/j.atmosres.2023.106933, 2023.

[7].Line 151: remove "the" of "the 85%"

**Response:**

Done. Thank you! **(Line 156)**

[8].Figure 1: How do the authors consider the elevation effect on ACI? In other words, do the authors consider the spatial variation in elevation height when studying ACI at each region?

**Response:**

Thank you for raising this point. Our study focuses on the regional-scale comparison between major, relatively flat megacity basins. While we acknowledge the potential influence of elevation, the complex terrain is not the primary focus of this work. We have added a sentence in Section 2.1 to acknowledge this limitation: "**It is noteworthy that the selected urban agglomerations are primarily situated in plains and basins, thus minimizing the potential confounding effects of**

**complex terrain on the aerosol-cloud-precipitation interactions analyzed in this study".** **(Lines 161-164)**

[9].Line 172-175: We should also note that hail also often occurs in spring or even summer in north China.

**Response:**

Thank you for raising this point. We agree that hail is a significant component of convective precipitation in spring or even summer in north China. Thus, we have revised the sentence to more accurately reflect the precipitation characteristics. "**This study focuses on liquid precipitation processes detectable by the GPM DPR during spring (April-May), summer (June-August), and autumn (September-November) from 2014 to 2023.**"(Lines 158-160)

Furthermore, to confine our analysis to liquid precipitation, we excluded any precipitation case that contained more than two consecutive pixels with the **flaghail** flag from the official GPM DPR products, thereby removing events likely dominated by hail.

[10].Line 183-188: How reliable is the aerosol classification?

**Response:**

Thank you for this critical question regarding the reliability of the aerosol classification. The aerosol classification in our study are derived from the Modern-Era Retrospective analysis for Research and Applications, Version 2 (MERRA-2) reanalysis product. MERRA-2 assimilates aerosol observations from multiple satellite sensors and ground-based AERONET measurements, which significantly constrains the model and improves the accuracy of its aerosol speciation (Buchard et al., 2017; Chang et al., 2015). The suitability of this dataset for investigating aerosol-precipitation interactions is demonstrated by its widespread adoption in related studies: "**Building on the reliable precipitation data from GPM DPR, researchers analyzed aerosol impacts on precipitation vertical structure, microphysical characteristics, and extreme hydrometeorological events using integrated MERRA–2 aerosol and DPR precipitation datasets (Ji and Tian, 2024; Jiang et al., 2023; Sun et al., 2022)**" (Lines 114-118)

In summary, while not without uncertainty, the MERRA-2 aerosol classification is a widely accepted and validated tool for large-scale climatological studies, providing a physically based and consistent framework for analyzing aerosol type contributions.

Buchard, V., Randles, C. A., Da Silva, A. M., Darmenov, A., Colarco, P. R., Govindaraju, R., Ferrare, R., Hair, J., Beyersdorf, A. J., Ziemba, L. D., and Yu, H.: The MERRA-2 aerosol reanalysis, 1980 onward. Part II: Evaluation and case studies, J. Climate, 30, 6851–6872,

https://doi.org/10.1175/JCLI-D-16-0613.1, 2017.

Chang, D., Cheng, Y., Reutter, P., Trentmann, J., Burrows, S. M., Spichtinger, P., Nordmann, S., Andreae, M. O., Pöschl, U., and Su, H.: Comprehensive mapping and characteristic regimes of aerosol effects on the formation and evolution of pyro-convective clouds, Atmos. Chem. Phys., 15, 10325–10348, https://doi.org/10.5194/acp-15-10325-2015, 2015.

Ji, Z. and Tian, S.: A novel potential cause of extreme precipitation in the northwest China, Heliyon, 10, e30826, https://doi.org/10.1016/j.heliyon.2024.e30826, 2024.

Jiang, M., Li, Y., Hu, W., Yang, Y., Brasseur, G., and Zhao, X.: Model-based insights into aerosol perturbation on pristine continental convective precipitation, Atmos. Chem. Phys., 23, 4545–4557, https://doi.org/10.5194/acp-23-4545-2023, 2023.

Sun, N., Fu, Y., Zhong, L., and Li, R.: Aerosol effects on the vertical structure of precipitation in east China, npj Clim Atmos Sci, 5, 60, https://doi.org/10.1038/s41612-022-00284-0, 2022.

[11].Line 189-190: How many hours are the aerosols used prior to precipitation events?

**Response:**

Thank you. In investigating aerosol impacts on precipitation, we employed the optimal spatiotemporal proximity matching principle to minimize interference.

Specifically, aerosol data from MERRA-2 grids were temporally and spatially matched to the nearest DPR precipitation pixels based on the timing of precipitation events. This approach uses aerosol observations closest to the onset of rainfall to approximate pre-precipitation aerosol conditions. Given that **"The data feature a global spatial coverage at a horizontal grid resolution of 0.625°×0.5° (longitude×latitude), with temporal products available at hourly intervals." (Lines 194-196)** Therefore, we utilized the MERRA-2 aerosol data from **within the hour** immediately preceding the observed precipitation event.

The revised text: **"It is noteworthy that the matched aerosol data are those within 1 hour prior to the onset of precipitation events." (Lines 196-197)**

[12].Line 191-198: similarly, the spatial and temporal resolution for data used here should be provided.

**Response:**

Done. Sorry for the careless. Thank you. The spatial and temporal resolutions for the ERA-5 (0.25°, 1h) data have been explicitly stated in Section 2.4. **" … … this product provides multidimensional climate parameters that span the surface-to-stratopause column spatial resolution of 0.25°×0.25° and temporal resolution of 1 hour (Hersbach et al., 2020)". (Lines 210-213)**

Hersbach, H., Bell, B., Berrisford, P., Hirahara, S., Horányi, A., Muñoz‑Sabater, J., Nicolas, J., Peubey, C., Radu, R., Schepers, D., Simmons, A., Soci, C., Abdalla, S., Abellan, X., Balsamo, G., Bechtold, P., Biavati, G., Bidlot, J., Bonavita, M., De Chiara, G., Dahlgren, P., Dee, D., Diamantakis, M., Dragani, R., Flemming, J., Forbes, R., Fuentes, M., Geer, A., Haimberger, L., Healy, S., Hogan, R. J., Hólm, E., Janisková, M., Keeley, S., Laloyaux, P., Lopez, P., Lupu, C., Radnoti, G., De Rosnay, P., Rozum, I., Vamborg, F., Villaume, S., and Thépaut, J.: The ERA5 global reanalysis, Quart J Royal Meteoro Soc, 146, 1999 – 2049, https://doi.org/10.1002/qj.3803, 2020.

[13].Line 197: A space is needed between the number and the unit: "850 hPa" instead of "850hPa" (also in Line 481).

**Response:**

Done. Thank you! **(Lines 214 and 523)**

[14].Line 204: The definition of "valid precipitation systems" requires further clarification. It is unclear whether the threshold of four contiguous precipitation pixels refers to nearsurface precipitation or to the existence of a precipitation profile. Some profiles may show precipitation aloft but none reaching the surface.

**Response:**

Thank you. We have clarified the definition in Section 2.5: **"Subsequently, precipitation pixels were screened using the connectivity method (Hu et al., 2022, 2024; Peng et al., 2025; Wang et al., 2024), applying a minimum threshold of four contiguous pixels (nsRR) to define valid precipitation systems (Liu and Zipser, 2015)." (Lines 217-220)**

Hu, X., Ai, W., Qiao, J., Hu, S., Han, D., and Yan, W.: Microphysics of Summer Precipitation Over Yangtze‑Huai River Valley Region in China Revealed by GPM DPR Observation, Earth and Space Science, 9, https://doi.org/10.1029/2021ea002021, 2022.

Hu, X., Ai, W., Qiao, J., and Yan, W.: Insight into global climatology of melting layer: Latitudinal dependence and orographic relief, Theor Appl Climatol, 155, 4863 – 4873, https://doi.org/10.1007/s00704-024-04926-6, 2024.

Liu, C. and Zipser, E. J.: The global distribution of largest, deepest, and most intense precipitation systems, Geophys. Res. Lett., 42, 3591 – 3595, https://doi.org/10.1002/2015gl063776, 2015.

Peng, H., Hu, X., Ai, W., Qiao, J., and Zhao, X.: Effects of fine and coarse aerosols on the summer precipitation structure and microphysics over the yangtze river delta region, Atmospheric Research, 326, 108277, https://doi.org/10.1016/j.atmosres.2025.108277, 2025.

Wang, Z., Hu, X., Ai, W., Qiao, J., and Zhao, X.: Microphysical characteristics of monsoon precipitation over yangtze-and-huai river basin and south China: A comparative study from GPM DPR observation, Remote Sens., 16, 3433, https://doi.org/10.3390/rs16183433, 2024.

[15].Line 208: "SO$_{4A}$" should be "SO$_4$A".

**Response:**

Done. Sorry for the careless error in the previous manuscript. It has been corrected in the manuscript. **(Line 229)**

[16].Line 232: Variable X should be defined.

**Response:**

Done. Thank you! The variable X has been defined in the revised manuscript: **" The X denote target precipitation parameter, which includes nsRR, STH, LWP, IWP, and precipitation efficiency index (PEI). The fractional changes (DIFFregion, in %) for each parameter in the YRD, YRM, and PRD regions relative to BTH were calculated as follows" (Lines 250-253)**

[17].Line 355-358: Relevant references should be added to support the statements.

**Response:**

Thank you for the good suggestions! Done. **"This seasonal contrast may be attributed to the abundant moisture supply during the pre−rainy season in South China (Chen and Luo, 2018), versus significant precipitation suppression in summer caused by hygroscopic aerosol −induced moisture competition (Guo et al.,2017) ."(Lines 385-388)**

Chen, Y. and Luo, Y.: Analysis of Paths and Sources of Moisture for the South China Rainfall during the Presummer Rainy Season of 1979 – 2014, J Meteorol Res, 32, 744 – 757, https://doi.org/10.1007/s13351-018-8069-7, 2018.

Guo, J., Su, T., Li, Z., Miao, Y., Li, J., Liu, H., Xu, H., Cribb, M., and Zhai, P.: Declining frequency of summertime local-scale precipitation over eastern China from 1970 to 2010 and its potential link to aerosols, Geophysical Research Letters, 44, 5700 – 5708, https://doi.org/10.1002/2017GL073533, 2017.

[18].Line 360: Fig. 5a mentioned here does not show RR and Ze; please check if this is a typographical error (should likely be "Fig. 4a").

**Response:**

Done. Sorry for the careless. Thank you. It has been corrected to **"Fig. 4a" (Line 380)**

[19].Lines 361-362: The statement "… Ze increase linearly with aerosol loading in the BTH …" seems valid only for the lower layers (<5 km) in the BTH according to Fig. 4b.

**Response:**

Thank you. Done. The sentence has been revised.: **"As shown in Fig. 4a-b, in spring, the RR and $Z_e$ for the lower layers (<5 km) increase linearly with aerosol loading in the BTH and YRD······"** (Lines 403-404)

[20].Line363-364: "Fig. 4a-e" may be a typographical error and should likely be "Fig. 3a-e".

**Response:**

Done. Sorry for the careless. Thank you. It has been corrected to "Fig. 3a-e" **(Line 406)**

[21].Line 365: "Fig. S1a" should be corrected to "Fig. S2a".

**Response:**

Done. Sorry for the careless. Thank you. It has been corrected to "Figs. S2-8" **(Line 407)**

[22].Line 364-367: It seems that the authors did not catch the finding from cited study, what they indicated is: This discrepancy may arise from the predominant influence of DUA on the spring AOD composition of the BTH region, whereby their radiative effect enhances the atmospheric instability and triggers earlier precipitation.

**Response:**

Thank you for point out this question. We apologize for the misalignment with the cited study. We have revised the sentence to accurately reflect the findings of Sun and Zhao (2021): **" In the BTH region, which is characterized by a high concentration of DUA(Figs. S2-8). Sun and Zhao (2021) attributed similar phenomena to the radiative effect of aerosols enhancing atmospheric instability and triggering earlier precipitation."** (Lines 406-409)

Sun, Y. and Zhao, C.: Distinct impacts on precipitation by aerosol radiative effect over three different megacity regions of eastern China, Atmos. Chem. Phys., 21, 16555–16574, https://doi.org/10.5194/acp-21-16555-2021, 2021.

[23].Lines 386–387: It might be helpful to clarify whether the term "particle" refers to aerosol particles or to cloud/raindrop particles, as this term appears in multiple places throughout the manuscript. A clear distinction may improve the overall clarity of the physical interpretation.

**Response:**

Thank you! We agree that the term "particle" was ambiguous. Thus, we have conducted a systematic review of the entire manuscript and clarified the term "particle" in all ambiguous contexts. The revisions are implemented as follows:

(1) In contexts discussing aerosol-cloud interactions, **"particle"** has been explicitly specified as **"aerosol particles"** (e.g., when referring to **CCN or IN**);

(2) In contexts discussing precipitation microphysics (e.g., within the coalescence and break-up analysis), "particle" has been explicitly specified as "cloud or raindrop particles". For instance, the **"Additionally, under low aerosol loadings, the lower atmosphere in the YRD (YRM) is dominated by high concentrations of smaller (larger) raindrop particles (Fig. 4c-d)." (Lines 429-431)**

Thank you again. This revision eliminates ambiguity and improves the clarity of physical interpretations throughout the manuscript.

[24].Figure 4: Unit for height should be given (also in Fig. 8).

**Response:**

Thank you for the reminder. Done! We have been updated all relevant figures **(Figs .4,9; Figs .S13,16-20).**

[25].Lines 403-407: The sentence could be simplified to avoid redundancy and improve clarity.

**Response:**

Thank you. The revised text: **"Consistent with Fig. 3k-o, during autumn the convective precipitation in the BTH and YRD initially decreases and then increases with rising aerosol loading (Fig. 4i-l). The magnitude of this increase is greater in the BTH than in the YRD." (Lines 446-448)**

[26].Line 410-413: The statement that prolonged cloud lifetime enhances precipitation should be supported with appropriate references and further discussion.

**Response:**

Thank you. Done! Specifically, we have added the foundational reference by Albrecht (1989) who first proposed the cloud lifetime effect, and key reviews by Rosenfeld et al. (2008) and Zhao et al. (2024, 2025) which provide comprehensive discussions on its role within the complex framework of aerosol-precipitation interactions. The revised sentence: **"The BTH and YRD: Initial aerosol increase elevates $N_w$ while reducing $D_m$ (Fig. 4k–l), suppressing precipitation via the Twomey effect. However, prolonged cloud lifetime promotes further cloud development (Fig. 3l–m; Albrecht 1989; Rosenfeld et al.,2008; Zhao et al.,2024,2025)." (Lines 450-453)**

Albrecht, B. A.: Aerosols, Cloud Microphysics, and Fractional Cloudiness, Science, 245, 1227－1230, https://doi.org/10.1126/science.245.4923.1227, 1989.

Rosenfeld, D., Lohmann, U., Raga, G. B., O＇Dowd, C. D., Kulmala, M., Fuzzi, S., Reissell, A., and Andreae, M. O.: Flood or Drought: How Do Aerosols Affect Precipitation?, Science, 321,

1309‒1313, https://doi.org/10.1126/science.1160606, 2008.

Zhao, C., Sun, Y., Yang, J., Li, J., Zhou, Y., Yang, Y., Fan, H., and Zhao, X.: Observational evidence and mechanisms of aerosol effects on precipitation, Science Bulletin, 69, 1569‒1580, https://doi.org/10.1016/j.scib.2024.03.014, 2024.

Zhao, X., Zhao, C., Chi, Y., Yang, J., Sun, Y., Yang, Y., and Fan, H.: Different Impacts of Aerosols on Cloud Development over Land and Ocean Regions in East China, Adv. Atmos. Sci., 42, 731‒743, https://doi.org/10.1007/s00376-024-4165-z, 2025.

[27].Lines 435–437: Relevant references should be added to support the statement.

**Response:**

Thank you this suggestion. We have added key citations to support the proposed microphysical mechanism. The revised text: **"This suggests that higher aerosol loads increase cloud albedo (Twomey et al., 1974; Garrett and Zhao, 2006), thereby intensifying the smaller cloud particles evaporation and break-up (Fig. 5b), resulting in a decline in PEI (Fig. S4j)."** **(Lines 477-480)**

Twomey, S.: Pollution and the planetary albedo, Atmospheric Environment (1967), 8, 1251‒1256, https://doi.org/10.1016/0004-6981(74)90004-3, 1974.

Garrett, T. J. and Zhao, C.: Increased Arctic cloud longwave emissivity associated with pollution from mid-latitudes, Nature, 440, 787‒789, https://doi.org/10.1038/nature04636, 2006.

[28].Line 453-455: I doubt if this claim is accurate or not. For example, there are almost no coalescence and break-up processes for non-precipitating clouds.

**Response:**

Thank you for this insightful comment. We agree that the original phrasing was overly broad and could be misinterpreted to apply to all clouds, including non-precipitating systems. The sentence has been revised to more precisely reflect the context of our study, which focuses on precipitating cloud systems. The revised text: **"The extensive application of this methodology in precipitating cloud systems demonstrates that coalescence and breakup processes are key mechanisms in cloud microphysics (Chen et al., 2025; Hu et al., 2022; Wen et al., 2023; Zhou et al., 2022). Consequently, this analysis focuses exclusively on coalescence and break-up mechanisms." (Lines 495-499)**

Chen, F., Yang, Y., Yu, L., Li, Y., Liu, W., Liu, Y., and Lolli, S.: Distinct effects of fine and coarse aerosols on microphysical processes of shallow-precipitation systems in summer over southern China, Atmos. Chem. Phys., 25, 1587‒1601, https://doi.org/10.5194/acp-25-1587-2025, 2025.

Hu, X., Ai, W., Qiao, J., Hu, S., Han, D., and Yan, W.: Microphysics of Summer Precipitation Over Yangtze‒Huai River Valley Region in China Revealed by GPM DPR Observation,

Earth and Space Science, 9, https://doi.org/10.1029/2021ea002021, 2022.

Wen, L., Chen, G., Yang, C., Zhang, H., and Fu, Z.: Seasonal variations in precipitation microphysics over East China based on GPM DPR observations, Atmospheric Research, 293, 106933, https://doi.org/10.1016/j.atmosres.2023.106933, 2023.

Zhou, L., Xu, G., Xiao, Y., Wan, R., Wang, J., and Leng, L.: Vertical structures of abrupt heavy rainfall events over southwest China with complex topography detected by dual‑frequency precipitation radar of global precipitation measurement satellite, Intl Journal of Climatology, 42, 7628–7647, https://doi.org/10.1002/joc.7669, 2022.

[29].Line 499: please rephrase the sentence to make it clear.

**Response:**

Thank you! We have rephrased the sentence to improve clarity: **"Under high RH during summer (Fig. 6f–j), the BTH response differs from that shown in Fig. 3f–j (characterized by a general increase in macroscopic precipitation parameters with aerosol loading); however, other regions remain consistent with prior trends in Fig. 3f–j." (Lines 552-555)**

[30].Section 5.2 and 5.3, please modify the section title to make it more reasonable.

**Response:**

Thank you for this suggestion regarding the section titles. We have revised them to more accurately and specifically reflect the content of each subsection: The updated titles are:

*5 Meteorological effects*

*5.1 Sensitivity of precipitation macroscopic parameters*

*5.2 Sensitivity of precipitation vertical profiles*

*5.3 Sensitivity of precipitation microphysical processes*

We sincerely appreciate your thorough review of our manuscript and the valuable feedback you provided. We have carefully addressed each comment and revised the manuscript accordingly, including additional clarifications on the methodology, refinements to data processing, deeper analysis of the results, and further validation of the conclusions. Your insightful suggestions have significantly enhanced the scientific rigor and logical coherence of this work, while also guiding future research directions. Thank you once again for your expert guidance and support. We look forward to your further feedback.

---

## Author Comment (AC2)

**Responses to Reviewers' Comments**

We are sincerely grateful to the editor and reviewers for their valuable time for reviewing our manuscript. The comments are very helpful and valuable, and we have addressed the issues raised by the reviewer in the revised manuscript. Please find our point-by-point response (in blue text) to the comments (in black text) raised by the reviewer. We have revised the paper according to your comments (**highlighted in blue text of the revised manuscript**).

**Reviewer #2:**

This manuscript focuses on aerosol-driven precipitation modification over four major megacity clusters in China (BTH, YRD, YRM, PRD), systematically analyzing the spatiotemporal heterogeneity of precipitation microphysics and vertical structures using integrated datasets such as GPM DPR, MERRA-2, and ERA5. The research topic is highly relevant to regional hydrological cycles and climate model refinement, as aerosol-precipitation interactions in densely populated, polluted urban agglomerations remain a key uncertaintyin atmospheric science. Overall, this manuscript is well organized: it investigates precipitation structural parameters, microphysical processes (e.g., coalescence, break-up), and the regulatory role of meteorological factors (RH, CAPE), providing a comprehensive framework for cross-regional comparisons. The conclusions regarding "regional precipitation disparities exceeding seasonal variations" and "aerosols mitigating regional differences in spring/summer" offer valuable insights for improving urban climate models. However, the manuscript has notable limitations that need to beaddressed, including insufficient quantification of aerosol type contributions, lack of analysis on joint meteorological factor interactions, and minor issues in details (e.g., figures). With appropriate modifications to strengthen mechanistic analysis and technical rigor, the manuscript will likely meet the publication standards.

**Response:**

Thank you for your thorough review and constructive feedback on our manuscript titled *"Aerosol-Driven Precipitation Modification: Spatiotemporal Heterogeneity in Precipitation Microphysics and Vertical Profiles over China's Megacity Clusters"*. We appreciate the time and effort you have invested in understanding our work and for providing us with clear directions for improvement. We confirm that all your earlier concerns were addressed in detail in our initial point-by-point response letter (submitted with the revised manuscript) and implemented in both the tracked-changes and clean versions of the manuscript. Below are our point-by-point responses:

**Major comments:**

[1]. Quantitative contribution of different types of aerosols to precipitation lacks clarity: The manuscript claims that BTH's convective precipitation is dominated by dust aerosols (semi-direct

effect), YRD/PRD by hygroscopic sea salt, and YRM by fine particles. However, it fails to provide quantitative data on the composition of aerosol types (DU, SS, SO₄, BC, OC from MERRA-2) across seasons and regions. For example:No temporal-spatial maps of aerosol type proportions (e.g., the proportion of dust column mass concentration in spring over BTH) are provided to support the "dust-dominated semi-direct effect" conclusion.

**Response:**

We thank you for this critical comment and fully agree that a more quantitative analysis is essential for attributing precipitation effects to specific aerosol species.

In sincere response to your valuable suggestion, we have significantly enhanced our manuscript to provide a more quantitative foundation. Specifically, we have supplemented our analysis with seasonal/regional distribution data of MERRA-2 aerosol type proportions/ and conducted correlation analyses between specific individual aerosol types and key precipitation parameters. These additions directly address your primary concerns and substantially strengthen the mechanistic interpretation of our results. **A detailed point-by-point explanation of how we have implemented these specific analyses is provided in our response to Comment #2 (Major) below, where we elaborate on the new tables and figures incorporated into the revised manuscript.**

We are profoundly grateful for the constructive comment again, which have been instrumental in improving the rigor and clarity of our work.

[2]. I strongly recommend the authors supplement (1) seasonal/regional distribution maps of MERRA-2 aerosol type proportions; (2) correlation analyses between individual aerosol types (e.g., DU in BTH, SS in PRD) and precipitation parameters (e.g., RR, Dm); (3) absorption aerosol optical depth (AAOD) data to distinguish the role of absorbing (BC, DU) vs. scattering (SO₄, SS) aerosols in microphysical processes. This will clarify the mechanism of aerosol type-specific impacts.

**Response:**

We sincerely thank you for these specific and constructive suggestions to quantitatively clarify the mechanisms of aerosol type-specific impacts. We fully agree that providing (1) seasonal/regional distribution maps of aerosol type proportions, (2) correlation analyses between individual aerosol types and precipitation parameters, and (3) utilizing AAOD and SAOD data to distinguish absorbing vs. scattering aerosols, would significantly strengthen our findings. In response, we have undertaken a comprehensive enhancement of our analysis:

(1) Seasonal/Regional Distribution of Aerosol Type Proportions and AOD Concentrations:

While our original analysis and SI did include some foundational data on aerosol type contributions (Fig S.2), we acknowledge that its presentation was insufficient to robustly support our mechanistic interpretations. Therefore, following your suggestion, we have now created and included detailed seasonal spatial distribution maps of both the average AOD and the proportional contributions of the seven MERRA-2 aerosol types (BC, DU, OC, SS, SO₄, absoring, and scattering) for each urban agglomeration (**now included as Figs. S3-S8 in the revised SI**).

The comparative analysis of both AOD concentration and proportional contribution maps reveals that the BTH region consistently exhibits the highest DUA burden among the four clusters (Figs. S3(b)-S8(b)), especially during spring. This regional specificity provides robust evidence for the proposed prominence of DUA and their associated semi-direct effect in shaping precipitation patterns in the BTH. Similarly, the figures show that SSA register the most pronounced signals in the PRD compared to the other regions (Figs. S3(d)-S8(d)), especially during spring and summer, firmly establishing the primary role in influencing local microphysics.

The corresponding figure citations and discussions for these findings have been incorporated into the main text of the revised manuscript. For instance:

**"The spatial and seasonal distributions of aerosol composition over China's megacity clusters, are detailed in Figs. S3-S8. Overall, the BTH region is characterized by a high burden of DUA (Figs. S3-S8b), particularly during spring. In contrast, the PRD region shows a pronounced signal of SSA, especially in spring and summer (Figs. S3-S8d). Figs. S3-S8 (f-g) show that loadings of both absorbing and scattering aerosols are markedly elevated in the BTH region, in contrast to the discernibly lower concentrations observed in the PRD. These characteristic distributions provide a fundamental basis for interpreting the region-specific aerosol-precipitation mechanisms discussed in subsequent sections"(Section 2.3, Lines 198-206)**

**"In the BTH region, which is characterized by a high concentration of DUA(Figs. S2-8). Sun and Zhao (2021) attributed similar phenomena to the radiative effect of aerosols enhancing atmospheric instability and triggering earlier precipitation." (Section 3.3, Lines 406-409)**

[Figure]

**Fig. S2.** Proportion of AOD in convective precipitation and stratiform precipitation under different

AOD loads in spring, summer, and autumn in the four regions.

**AOD-Spring (2014-2023)**

[Figure]

**Fig. S3.** Spatial distribution map of seven types (BC, DU, OC, SS, SO$_4$, absoring, and scattering) of average aerosols in spring from 2014 to 2023 over China's Megacity Clusters ( BTH, YRD, YRM, and PRD).

**AOD Ratio-Spring (2014-2023)**

[Figure]

**Fig. S4.** Spatial distribution map of seven types (BC, DU, OC, SS, SO$_4$, absoring, and scattering) of aerosol type proportions in spring from 2014 to 2023 over China's Megacity Clusters (BTH, YRD, YRM, and PRD).

[Figure]

**Fig. S5.** Spatial distribution map of seven types (BC, DU, OC, SS, SO₄, absoring, and scattering) of average aerosols in summer from 2014 to 2023 over China's Megacity Clusters ( BTH, YRD, YRM, and PRD).

[Figure]

**Fig. S6.** Spatial distribution map of seven types (BC, DU, OC, SS, SO₄, absoring, and scattering) of aerosol type proportions in summer from 2014 to 2023 over China's Megacity Clusters (BTH, YRD, YRM, and PRD).

[Figure]

**Fig. S7.** Spatial distribution map of seven types (BC, DU, OC, SS, SO₄, absoring, and scattering) of average aerosols in autumn from 2014 to 2023 over China's Megacity Clusters ( BTH, YRD, YRM, and PRD).

[Figure]

**Fig. S8.** Spatial distribution map of seven types (BC, DU, OC, SS, SO₄, absoring, and scattering) of aerosol type proportions in autumn from 2014 to 2023 over China's Megacity Clusters (BTH, YRD, YRM, and PRD).

(2)  Correlation Analyses between Individual Aerosol Types and Precipitation Parameters:

We have conducted and now present in the revised manuscript (Section 3.3) the results of Spearman correlation analyses between the concentrations of key aerosol types (DU in BTH, SS in PRD) and critical precipitation parameters (RR, $D_m$, and $N_w$). Specifically, for RR, $N_w$ and $D_m$, we adopted the mean values within the 1–3 km altitude range as the core indicators.

The newly added correlation analyses figure (Fig. S12) further elucidate the distinct seasonal and precipitation type dependent roles of key aerosol species. We have added the sentence to specify that: **"Despite a complex influence that varies with season and precipitation type, DUA in the BTH region exhibit a consistent negative correlation with $N_w$ (Fig. S12b-e). This negative correlation could be attributed to the semi-direct effect of DUA, whereby the absorption of solar radiation heats the atmosphere, thus promoting cloud droplet evaporation and suppressing droplet concentrations. The negative relationship with RR and $D_m$ is particularly evident in stratiform precipitation under low-to-medium aerosol loadings (Fig. S12d). For convective precipitation, the influence of DUA is seasonally modulated, showing a negative correlation in summer but a positive correlation in spring and autumn. Meanwhile, in the PRD region (Fig. S12g-l), SSA during spring and summer are consistently positively correlated with precipitation parameters. These statistically significant relationships further corroborate the proposed type-specific mechanisms."** (Section 3.3. Lines 391-402)

For completeness, we conducted an extensive analysis encompassing all five individual aerosol species **(below the Fig. S12)**. To maintain clarity and conciseness in the main narrative, we have presented the analysis for the most regionally representative types as a paradigm.

[Figure]

Fig. S12. Spearman correlation coefficients between key aerosol types (DUA for the BTH and SSA for the PRD) and precipitation parameters (RR, $N_w$ and $D_m$) across regions and seasons under the three AOD regimes. Color gradients (from yellow to blue) encode the correlation strength and direction, and asterisks denote statistical significance (*: $p < 0.05$, **: $p < 0.01$, ***: $p < 0.001$).

[Figure]

Furthermore, in direct response to a related point raised by Reviewer Comment#1: *"While the authors emphasize the importance of the semi-direct effect of dust in the BTH region, dust is also a major source of ice nuclei, which could substantially influence cold-toppedprecipitation formation."*, we have conducted an additional analysis to specifically investigate the role of DU as ice nuclei (IN) in the BTH region. This involved examining the correlations between DUA and both the Ice Water Path (IWP) and Storm Top Height (STH).

The results reveal a notable dual role of dust aerosols: **"Given the dominant roles of DUA in BTH and SSA in PRD (Figs. S2-S8), this study further investigated how these key aerosol types distinctly modulate precipitation. Correlation analyses reveal that DUA is significantly positively correlated with both IWP and STH in the convective precipitation (Fig. S10), indicating an invigoration of ice-phase processes. This suggests that DUA, acting as efficient IN, promote the glaciation and vertical development of convective clouds, consistent with the invigoration effect (Rosenfeld., 2008). In contrast, the impact of DUA on stratiform precipitation is substantially weaker and more variable. This promotional mechanism, facilitated by ice-nucleating ability of DUA, stands in contrast to precipitation-suppressing semi-direct effect, highlighting the complex and multi-faceted nature of DUA impacts on different precipitation types." (Section 3.1. Lines 306-316)**

[Figure]

**Fig. S10.** Spearman correlation coefficients between DUA and precipitation parameters (STH, and IWP) in BTH for convective and stratiform precipitation under three AOD regimes and seasons.

(3) Consideration of Absorbing Aerosols and Scattering Aerosols:

We deeply appreciate the insightful recommendation to use Absorption Aerosol Optical Depth (AAOD) and Scattering Aerosol Optical Depth (SAOD). Firstly, we categorized the aerosols as absorbing (BC, DU) and scattering (SO$_4$, SS) based on their fundamental radiative properties (Section 2.3). Secondly, detailed seasonal spatial distribution maps of both the average AOD and the proportional contributions are now visualized in the new supplement information(Figs. S3-S8). **"Figs. S3-S8 (f-g) show that loadings of both absorbing and scattering aerosols are markedly elevated in the BTH region, in contrast to the discernibly lower concentrations observed in the PRD." (Section 2.3, Lines 202-204)** Finally, we explicitly attribute the observed precipitation modifications to the fundamental processes associated with specific aerosol types and conditions. The revised manuscript highlights the following physical mechanisms through which aerosols modify precipitation:

❖ The semi-direct effect for strongly absorbing aerosols (e.g., dust in BTH).

❖ The Twomey and cloud lifetime effects for predominantly scattering aerosols (e.g., sulfate).

❖ The convective invigoration effect, particularly for dust aerosols acting as efficient IN, which enhances ice-phase processes and deepens convective clouds.

❖ The hygroscopic growth and coalescence-enhancement effect associated with coarse-mode aerosols (e.g., sea salt in PRD).

In summary, your comments have been instrumental in driving a significant upgrade to our manuscript. The new figures and correlation analyses provide the clear, quantitative evidence needed to solidify our conclusions regarding aerosol type-specific impacts.

Rosenfeld, D., Lohmann, U., Raga, G. B., O'Dowd, C. D., Kulmala, M., Fuzzi, S., Reissell, A., and Andreae, M. O.: Flood or Drought: How Do Aerosols Affect Precipitation?, Science, 321, 1309–1313, https://doi.org/10.1126/science.1160606, 2008.

Sun, Y. and Zhao, C.: Distinct impacts on precipitation by aerosol radiative effect over three different megacity regions of eastern China, Atmos. Chem. Phys., 21, 16555–16574, https://doi.org/10.5194/acp-21-16555-2021, 2021.

[3].Ambiguity in precipitation type classification: The manuscript excludes "shallow convection" from convective precipitation based on 2ADPR criteria but does not specify the 2ADPR threshold for shallow convection and report the proportion of shallow convection in total precipitation across regions/seasons. This will affect sample representativeness and undermine the robustness of the findings.

**Response:**

We thank you for the insightful comment regarding the precipitation type classification. We agree that clarity on this point is crucial for the robustness of our findings. We provide a detailed clarification of our methodology, which is firmly based on the GPM DPR (2ADPR V07) Algorithm Theoretical Basis Document (ATBD).

(1) Definition of Precipitation Based on GPM DPR Criteria:

In our study, precipitation types were classified using GPM DPR products (2ADPR V07). Shallow convection was identified as a subset of the convective precipitation using the following criterion:

$$index\_Shallow = (heightZeroAll - STHAll > 1) \ \& \ (typeAll == 2)$$

Similarly, the classification of the stratiform and convective precipitation in our study were as follows:

$$inde\_Stra = (STHAll > heightZeroAll) \ \& \ (typeAll == 1)$$

$$index\_Conv = (STHAll > heightZeroAll) \ \& \ (typeAll == 2)$$

Here, h*eightZeroAll−STHAll* > 1 indicates that the storm top is more than 1 km below the freezing level,which threshold is consistent with the ATBD (See figure below). The typeAll is the precipitation type flag from 2ADPR, where 1 denotes stratiform and 2 denotes convective. The condition *STHAll* > h*eightZeroAll* was applied to both types to ensure we focused on sufficiently developed precipitating systems.
* * *
(c) **Shallow rain and small cell-size rain**

Detection of shallow rain is also made independently of the above mentioned methods of rain type classification. When the following condition is satisfied, it is judged as shallow rain, which will be marked by an internal flag:

$$\texttt{heightStormTop} < \texttt{heightZeroDeg} - \text{margin}$$

where margin is currently 1000 m.

Shallow rain is separated out into shallow isolated and shallow non-isolated by examining the horizontal extent of shallow rain. In the rain type unification, both shallow isolated and shallow non-isolated are classified as convective. (It should be noted that in the TRMM PR rain type classification algorithm 2A23 V7, all the shallow isolated is convective, but shallow non-isolated can be either stratiform or convective.)

Detection of rain having small cell size is also made independently. The rain having small cell size is classified as convective in the unification of rain type.
* * *
Definition and Classification of Shallow Rain in the GPM DPR Algorithm.
(https://gpm.nasa.gov/resources/documents/gpmdpr-level-2-algorithm-theoretical-basis-document-atbd)

(2) Proportion of Shallow Convection in Total Precipitation:

As suggested, we have calculated the seasonal and regional proportions of shallow convection relative to total convective precipitation. The results are summarized below (as a percentage of total precipitation events):

Table. S1 Shallow convection statistics: event counts and percentage contributions to convective and total precipitation (analyzed in this study) across regions and seasons

| Region | Season | Shallow Counts | Shallow/Conv. (%) | Shallow/Total (%) |
|--------|--------|----------------|-------------------|-------------------|
| BTH | Spring | 570 | 13.60 | 1.01 |
| | Summer | 7860 | 30.27 | 5.83 |

| | | | | |
|---|---|---|---|---|
| | Autumn | 1904 | 44.95 | 5.23 |
| | Spring | 9575 | 57.32 | 8.62 |
| YRD | Summer | 23268 | 53.74 | 13.92 |
| | Autumn | 4860 | 68.78 | 14.59 |
| | Spring | 23945 | 41.69 | 6.78 |
| YRM | Summer | 63536 | 51.58 | 13.16 |
| | Autumn | 18959 | 69.89 | 13.23 |
| | Spring | 2765 | 37.08 | 9.47 |
| PRD | Summer | 6961 | 46.29 | 20.15 |
| | Autumn | 2349 | 49.02 | 17.19 |

These proportions reveal significant regional and seasonal variations in the shallow convection precipitation. While shallow convection represents a relatively small fraction in some regions and seasons (e.g., 13.60% in BTH Spring), it constitutes a substantial portion of total convective precipitation in others (e.g., 69.89% in YRM Autumn). This heterogeneity underscores the importance of our methodological approach in excluding shallow convection to ensure physical consistency across our analysis. Importantly, the convective sample sizes across all regions and seasons exceed 2,000 events, providing robust statistical power for our analyses.

We have revised the manuscript to include a clear specification of the classification criteria for all precipitation types, including the shallow convection threshold, in the Data and Methods section. **"This study categorized precipitation into stratiform and convective types based on the 2ADPR classification criteria, excluding shallow convection events (defined by the 0℃ isotherm attitude below STH more than 1 km) from convective precipitation (Liu and Zipser, 2015). It is noteworthy that shallow convection precipitation constitutes a non-negligible proportion of the convective events in our dataset (Table S1). This prevalence underscores the necessity of our methodological decision to exclude these events. " (Section 2.5. Lines 220-226)**

We believe this clarification fully addresses your concern and underscores the methodological rigor of our study.

Liu, C. and Zipser, E. J.: The global distribution of largest, deepest, and most intense precipitation systems, Geophysical Research Letters, 42, 3591–3595, https://doi.org/10.1002/2015gl063776, 2015.

[4].It is well acknowledged that favorable meteorological conditions are indispensable for the formation and evolution of precipitation, thereby inevitably making it elusive to disentangle the aerosol effect on precipitation. This manuscript analyzes RH (thermodynamic) and CAPE (dynamic) separately but ignores their synergistic or antagonistic effects on aerosol-precipitation interactions. The readers are more willing to see does aerosol-induced coalescence strengthen more than in single-factor conditions under the high RH (sufficient moisture) + high CAPE (strong updrafts) conditions. The authors may conduct a two-factor crossed analysis (e.g., 3 RH levels × 3 CAPE levels) to quantify aerosol impacts on precipitation parameters under different

combined meteorological scenarios. This will reveal the regulatory mechanism of thermodynamic-dynamic synergy, improving the comprehensiveness of the conclusion.

**Response:**

We sincerely thank you for this insightful comment regarding the potential synergistic or antagonistic effects of meteorological factors on aerosol-precipitation interactions. We fully agree that a two-factor crossed analysis could provide a more nuanced understanding of how thermodynamic and dynamic conditions jointly modulate aerosol impacts.

However, conducting a comprehensive two-factor crossed analysis (3 RH × 3 CAPE) within the existing framework (which already encompasses four regions, three seasons, two precipitation types and three AOD levels), would lead to a substantial fragmentation of the sample size into numerous sub-groups. Many of these sub-groups, especially those representing specific combinations of high RH and high CAPE under certain AOD regimes in particular regions and seasons, would contain an insufficient number of samples. This would severely compromise the statistical significance and robustness of the results, potentially introducing considerable uncertainty rather than providing clear mechanistic insights. The bubble chart below depicts the sample distribution across these RH-CAPE combinations (The classification of RH and CAPE values in the figure is consistent with the three conditions used in the original manuscript, though without further subdivision by precipitation type or AOD level):

**"As shown in Fig. 6 a–d, an increase in RH generally enhances the near-surface rain rate (nsRR), and the combination of high RH × high CAPE (indicated by ☆) tends to produce heavier precipitation. However, as the CAPE values (Fig. 6 e–h) continue to rise beyond 4000 J/kg, the nsRR shows a decreasing trend. Moreover, extremely high CAPE values are most often observed alongside moderate to low RH (indicated by ○ and △) conditions." (Section 5.1. Lines 527-532)**

[Figure]

**Fig. 6.** nsRR as a function of RH-850 hPa (CAPE), stratified by season (color) and CAPE (RH) level (symbol), for all precipitation types (convective and stratiform) in the four regions. The symbol size is proportional to the sample size ([0,20), [20,45), [45,90), [90,~)).

In our current study, we have systematically analyzed the individual influences of RH and CAPE across different regions, seasons, and aerosol loadings (in Section 5). This approach has already yielded significant insights. For instance, we found that elevated RH generally promotes coalescence and enhances precipitation parameters, whereas increased CAPE can enhance cloud development but also intensify break-up processes, suppressing precipitation under certain conditions.

Therefore, while we acknowledge the limitation of not performing the specific synergistic analysis suggested, we believe that our detailed separate analyses of RH and CAPE, combined with the investigation of microphysical processes, have effectively captured the primary individual and indirect roles of these key meteorological factors. We have explicitly acknowledged this aspect as a scope for future work in the revised Discussion section: **"Additionally, a more complete understanding of aerosol-precipitation interactions must account for the complex synergies among environmental factors, leveraging advanced statistical or modeling methods in a multi-factor analytical framework." (Section 7. Lines 793-796)**

We are grateful for this valuable suggestion again, which provides a clear and important direction for our subsequent researches.

[5].In Results and discussion part: I recommend adding one or two paragraphs to focus on key findings and providing a comparative discussion on how these findings align with or diverge from previous studies on aerosol-precipitation interaction. This will help more effectively highlight the study's unique contributions to the community.

**Response:**

We sincerely thank you for this insightful suggestion. Therefore, we have now added dedicated paragraph in the Results and Discussion section to synthesize the key findings and explicitly compare them with previous studies:

**"Building on the findings of Peng et al. (2025), which investigated the effects of fine and coarse aerosols on summer precipitation structure and microphysics in the YRD region, the present study expands the scope of analysis to examine aerosol impacts on precipitation vertical profiles and microphysical processes across multiple regions and seasons in China. This expanded scope, combined with a unified analytical methodology, enables a systematic cross-regional and cross-seasonal comparison that mitigates inconsistencies often associated with disparate data sources or methods, yielding the following key findings: (1)Enhanced aerosol loading reduces regional precipitation disparities, most pronounced in spring and summer. (2)Precipitation exhibits stronger regional than seasonal variability. (3)The BTH precipitation is dominated by dust aerosols, whereas the YRD and PRD are influenced by sea salt aerosols. These conclusions are primarily derived from analyses of satellite-based datasets, which provide extensive spatial coverage, high spatiotemporal resolution, and continuous temporal monitoring." (Section 7. Lines 743-756)**

**"The results provide further evidence for several established mechanisms: The dominant role of dust aerosols in the BTH region aligns with existing research ( Xi et al.,**

**2024; Xiao et al.,2025), manifesting as impacts on precipitation through ARI (Sun and Zhao, 2022) and semi-direct effects, and also serving as effective IN to promote convective cloud development (Xi et al.,2025). Furthermore, the precipitation-enhancing effect of sea salt aerosols in the PRD region is consistent with prior observations (Guo et al., 2022; Chen et al.,2025). (Section 7. Lines 757-763)**

**However, as noted by Stier et al. (2024) and Zhao et al. (2025), aerosol impacts on precipitation remain highly complex, and their net effects are still subject to considerable uncertainty across different scales. Multiple factors are known to modulate precipitation processes, such as vertical wind shear (Kim et al., 2003; Guo et al.,2018), cloud properties (Shao and Liu, 2005; Zhao et al., 2012), and latent heating (Zhu et al., 2025). Therefore, a critical challenge for future research lies in better disentangling the influence of such environmental meteorological factors from the overall aerosol effect." (Section 7. Lines 764-771)**

We believe these additions significantly strengthen the discussion by directly addressing the interplay between our new findings and the established body of knowledge, thereby clarifying unique value of our research. We are grateful for this constructive suggestion.

Chen, F., Yang, Y., Yu, L., Li, Y., Liu, W., Liu, Y., and Lolli, S.: Distinct effects of fine and coarse aerosols on microphysical processes of shallow-precipitation systems in summer over southern China, Atmos. Chem. Phys., 25, 1587–1601, https://doi.org/10.10194/acp-25-1587-2025, 2025.

Guo, J., Luo, Y., Yang, J., Furtado, K., and Lei, H.: Effects of anthropogenic and sea salt aerosols on a heavy rainfall event during the early-summer rainy season over coastal Southern China, Atmospheric Research, 265, 105923, https://doi.org/10.1016/j.atmosres.2021.105923, 2022.

Guo, J., Liu, H., Li, Z., Rosenfeld, D., Jiang, M., Xu, W., Jiang, J. H., He, J., Chen, D., Min, M., and Zhai, P.: Aerosol-induced changes in the vertical structure of precipitation: a perspective of TRMM precipitation radar, Atmos. Chem. Phys., 18, 13329–13343, https://doi.org/10.5194/acp-18-13329-2018, 2018.

Peng, H., Hu, X., Ai, W., Qiao, J., and Zhao, X.: Effects of fine and coarse aerosols on the summer precipitation structure and microphysics over the yangtze river delta region, Atmospheric Research, 326, 108277, https://doi.org/10.1016/j.atmosres.2025.108277, 2025.

Shao, H. and Liu, G.: Why is the satellite observed aerosol's indirect effect so variable?, Geophysical Research Letters, 32, https://doi.org/10.1029/2005GL023260, 2005.

Stier, P., van den Heever, S. C., Christensen, M. W., Gryspeerdt, E., Dagan, G., Saleeby, S. M., Bollasina, M., Donner, L., Emanuel, K., Ekman, A. M. L., Feingold, G., Field, P., Forster, P., Haywood, J., Kahn, R., Koren, I., Kummerow, C., L'Ecuyer, T., Lohmann, U., Ming, Y., Myhre, G., Quaas, J., Rosenfeld, D., Samset, B., Seifert, A., Stephens, G., and Tao, W.-K.: Multifaceted aerosol effects on precipitation, Nat. Geosci., 17, 719–732, https://doi.org/10.1038/s41561-024-01482-6, 2024.

Sun, Y., Wang, Y., Zhao, C., Zhou, Y., Yang, Y., Yang, X., Fan, H., Zhao, X., and Yang, J.: Vertical Dependency of Aerosol Impacts on Local Scale Convective Precipitation, Geophysical Research Letters, 50, https://doi.org/10.1029/2022gl102186, 2023.

Sun, Y. and Zhao, C.: Distinct impacts on precipitation by aerosol radiative effect over three

different megacity regions of eastern China, Atmos. Chem. Phys., 21, 16555–16574, https://doi.org/10.5194/acp-21-16555-2021, 2021.

Xi, J., Li, R., Fan, X., and Wang, Y.: Aerosol effects on the three-dimensional structure of organized precipitation systems over Beijing-Tianjin-Hebei region in summer, Atmospheric Research, 298, 107146, https://doi.org/10.1016/j.atmosres.2023.107146, 2024.

Xi, J., Wang, Y., Li, R., Wu, B., Fan, X., Ma, X., and Meng, Z.: The impact of Sahara dust aerosols on the three-dimensional structure of precipitation systems of different sizes in spring, EGUsphere, 1–30, https://doi.org/10.5194/egusphere-2025-2799, 2025.

Xiao, Y., Zhang, J., Zhu, J., and Dai, Q.: Exploration of aerosol-precipitation relationships under different climate regimes in China, GIScience & Remote Sensing, 62, https://doi.org/10.1080/15481603.2025.2457992, 2025.

Zhao, C., Klein, S. A., Xie, S., Liu, X., Boyle, J. S., and Zhang, Y.: Aerosol first indirect effects on non-precipitating low-level liquid cloud properties as simulated by CAM5 at ARM sites, Geophysical Research Letters, 39, https://doi.org/10.1029/2012GL051213, 2012.

Zhao, X., Zhao, C., Chi, Y., Yang, J., Sun, Y., Yang, Y., and Fan, H.: Different Impacts of Aerosols on Cloud Development over Land and Ocean Regions in East China, Adv. Atmos. Sci., 42, 731–743, https://doi.org/10.1007/s00376-024-4165-z, 2025.

Zhu, H., Zhao, H., Yang, S., Zhou, R., Wang, Y., Zou, Y., Zhao, C., and Li, R.: Smoke aerosols elevate precipitation top and latent heat to the upper atmosphere globally, npj Clim Atmos Sci, 8, https://doi.org/10.1038/s41612-025-01047-3, 2025.

**Minor comments**

[1] Lines 71-73: Except for the external synoptic conditions that can modulate the ACI process, entrainment, and other in-cloud meteorological factors, particularly that surrounding clouds and challenging to be measured, can affect the aerosol effect on clouds and precipitation.

**Response:**

We thank you for this insightful comment. In the revised manuscript, we have acknowledged this point in the Introduction to provide a more comprehensive background: "**Furthermore, external synoptic conditions, entrainment, and other in-cloud meteorological factors, particularly that surrounding clouds and challenging to be measured, can affect the aerosol effect on clouds and precipitation (Lee et al.,2008; Stevens and Feingold,2009; Chen et al., 2025; Sun et al., 2023; Zhao et al., 2024)**. " **(Lines 71-75)**

Chen, F., Yang, Y., Yu, L., Li, Y., Liu, W., Liu, Y., and Lolli, S.: Distinct effects of fine and coarse aerosols on microphysical processes of shallow-precipitation systems in summer over southern China, Atmos. Chem. Phys., 25, 1587–1601, https://doi.org/10.5194/acp-25-1587-2025, 2025.

Lee, S. S., Donner, L. J., Phillips, V. T. J., and Ming, Y.: The dependence of aerosol effects on clouds and precipitation on cloud-system organization, shear and stability, Journal of Geophysical Research: Atmospheres, 113, https://doi.org/10.1029/2007JD009224, 2008.

Stevens, B. and Feingold, G.: Untangling aerosol effects on clouds and precipitation in a buffered
system, Nature, 461, 607–613, https://doi.org/10.1038/nature08281, 2009.

Sun, Y., Wang, Y., Zhao, C., Zhou, Y., Yang, Y., Yang, X., Fan, H., Zhao, X., and Yang, J.: Vertical
Dependency of Aerosol Impacts on Local Scale Convective Precipitation, Geophysical
Research Letters, 50, https://doi.org/10.1029/2022gl102186, 2023.

Zhao, C., Sun, Y., Yang, J., Li, J., Zhou, Y., Yang, Y., Fan, H., and Zhao, X.: Observational
evidence and mechanisms of aerosol effects on precipitation, Science Bulletin, 69,
1569–1580, https://doi.org/10.1016/j.scib.2024.03.014, 2024.

[2]. Line 75: the citations are not correctly placed. Actually, these references are used to support
"Significant research in recent years has focused on aerosol–induced modifications of
precipitation structures in key regions of China".

**Response:**

Sorry for the careless error in the previous manuscript. Thank you! We have corrected the
citation placement to properly support the statement on recent research in key Chinese regions.
**"Recent research has extensively examined aerosol-induced modifications of precipitation
structures in major urban clusters across China (Guo et al., 2018; Sun and Zhao, 2021; Zhao
et al., 2025)." (Lines 76-78)**

Guo, J., Liu, H., Li, Z., Rosenfeld, D., Jiang, M., Xu, W., Jiang, J. H., He, J., Chen, D., Min, M.,
and Zhai, P.: Aerosol-induced changes in the vertical structure of precipitation: a perspective
of TRMM precipitation radar, Atmos. Chem. Phys., 18, 13329–13343,
https://doi.org/10.5194/acp-18-13329-2018, 2018.

Sun, Y. and Zhao, C.: Distinct impacts on precipitation by aerosol radiative effect over three
different megacity regions of eastern China, Atmos. Chem. Phys., 21, 16555–16574,
https://doi.org/10.5194/acp-21-16555-2021, 2021.

Zhao, X., Zhao, C., Chi, Y., Yang, J., Sun, Y., Yang, Y., and Fan, H.: Different Impacts of Aerosols
on Cloud Development over Land and Ocean Regions in East China, Adv. Atmos. Sci., 42,
731–743, https://doi.org/10.1007/s00376-024-4165-z, 2025.

[3]. The argument "analysis of specific seasons or precipitation types is frequently limited" is not
correct. To my knowledge, the following references have investigated the aerosol effect on
precipitation types, such as https://doi.org/10.1073/pnas.1715386115; https://doi.
org/10.1029/2019GL085442

**Response:**

We sincerely thank you for correcting this argument and for providing the relevant literature.
We have revised the statement in the Introduction (Lines 82-85) to more accurately reflect the
existing research landscape: **"Previous research has been conducted on aerosol-driven
precipitation patterns in more general areas, such as the North China Plain (Sun et al., 2023),**

**South China (Chen et al., 2025), and East China (Wen et al., 2023) under different seasonal and precipitation type conditions (Day et al., 2018; Guo et al., 2019). These studies have consistently confirmed that the modulation effects of aerosols on clouds and precipitation exhibit pronounced regional heterogeneity (Guo et al., 2017, 2019; Li et al., 2019; Sun and Zhao, 2021). However, the underlying physical mechanisms, particularly the cloud microphysical processes responsible for these disparate regional responses, are not yet fully understood." (Lines 82-90)**

Chen, F., Yang, Y., Yu, L., Li, Y., Liu, W., Liu, Y., and Lolli, S.: Distinct effects of fine and coarse aerosols on microphysical processes of shallow-precipitation systems in summer over southern China, Atmos. Chem. Phys., 25, 1587–1601, https://doi.org/10.5194/acp-25-1587-2025, 2025.

Day, J. A., Fung, I., and Liu, W.: Changing character of rainfall in eastern China, 1951–2007, Proceedings of the National Academy of Sciences, 115, 2016–2021, https://doi.org/10.1073/pnas.1715386115, 2018.

Guo, J., Su, T., Li, Z., Miao, Y., Li, J., Liu, H., Xu, H., Cribb, M., and Zhai, P.: Declining frequency of summertime local-scale precipitation over eastern China from 1970 to 2010 and its potential link to aerosols, Geophysical Research Letters, 44, 5700–5708, https://doi.org/10.1002/2017GL073533, 2017.

Guo, J., Su, T., Chen, D., Wang, J., Li, Z., Lv, Y., Guo, X., Liu, H., Cribb, M., and Zhai, P.: Declining Summertime Local-Scale Precipitation Frequency Over China and the United States, 1981–2012: The Disparate Roles of Aerosols, Geophysical Research Letters, 46, 13281–13289, https://doi.org/10.1029/2019GL085442, 2019.

Li, Z., Wang, Y., Guo, J., Zhao, C., Cribb, M. C., Dong, X., Fan, J., Gong, D., Huang, J., Jiang, M., Jiang, Y., Lee, S.-S., Li, H., Li, J., Liu, J., Qian, Y., Rosenfeld, D., Shan, S., Sun, Y., Wang, H., Xin, J., Yan, X., Yang, X., Yang, X., Zhang, F., and Zheng, Y.: East asian study of tropospheric aerosols and their impact on regional clouds, precipitation, and climate (EAST-AIRCPC), Journal of Geophysical Research: Atmospheres, 124, 13026–13054, https://doi.org/10.1029/2019JD030758, 2019.

Sun, Y. and Zhao, C.: Distinct impacts on precipitation by aerosol radiative effect over three different megacity regions of eastern China, Atmos. Chem. Phys., 21, 16555–16574, https://doi.org/10.5194/acp-21-16555-2021, 2021.

Sun, Y., Wang, Y., Zhao, C., Zhou, Y., Yang, Y., Yang, X., Fan, H., Zhao, X., and Yang, J.: Vertical Dependency of Aerosol Impacts on Local Scale Convective Precipitation, Geophysical Research Letters, 50, https://doi.org/10.1029/2022gl102186, 2023.

Wen, L., Chen, G., Yang, C., Zhang, H., and Fu, Z.: Seasonal variations in precipitation microphysics over East China based on GPM DPR observations, Atmospheric Research, 293, 106933, https://doi.org/10.1016/j.atmosres.2023.106933, 2023.

[4]. "Aerosol loading" and "aerosol concentration" are used interchangeably. The authors may unify to one terminology. And "vertical structures" and "vertical profiles" (used for Ze, Dm) can be unified to "vertical profiles" for consistency.

**Response**

We appreciate this suggestion for improving terminological consistency. We have revised the manuscript accordingly: using **"aerosol loading"** and **"vertical profiles"** consistently.

[5]. The discussion mentions EarthCARE's potential for aerosol-cloud vertical profiling but does not specify how its data (ATLID lidar aerosol profiles, CPR cloud profiles) will address the current study's limitation of "inadequate 3D aerosol matching". I recommend adding 1–2 sentences on future research directions (e.g., combining EarthCARE's aerosol vertical distribution with GPM DPR's precipitation profiles to analyze aerosol-altitude-dependent impacts on cloud microphysics), enhancing the discussion's innovations.

**Response:**

This is an excellent suggestion. Thank you! We have expanded the Discussion to specify the future application of EarthCARE data: **"However, the successful launch and stable operation of EarthCARE now facilitates accurate three-dimensional vertical profiling of clouds and aerosols via lidar (ATLID) and cloud profiling radar (CPR) (Irbah et al., 2023). The integration of high-precision vertical profiles from ATLID (aerosols), CPR (clouds), and GPM DPR (precipitation) will enable future researchers to quantify how aerosol layers at different altitudes modulate cloud microphysics and precipitation formation (Li et al., 2025b)." (Lines 781-787)**

Irbah, A., Delanoë, J., Van Zadelhoff, G.-J., Donovan, D. P., Kollias, P., Puigdomènech Treserras, B., Mason, S., Hogan, R. J., and Tatarevic, A.: The classification of atmospheric hydrometeors and aerosols from the EarthCARE radar and lidar: the A-TC, C-TC and AC-TC products, Atmos. Meas. Tech., 16, 2795–2820, https://doi.org/10.5194/amt-16-2795-2023, 2023.

Li, Z., Ge, S., Hu, X., Ai, W., Tang, J., Qiao, J., Hu, S., Zhao, X., Wu, H., Li, Z., Ge, S., Hu, X., Ai, W., Tang, J., Qiao, J., Hu, S., Zhao, X., and Wu, H.: Preliminary analysis of a novel spaceborne pseudo tripe-frequency radar observations on cloud and precipitation: EarthCARE CPR-GPM DPR coincidence dataset, Remote Sens., 17, https://doi.org/10.3390/rs17152550, 2025b.

[6]. Figure 1 caption can be rephrased as "Geographical distribution of four urban agglomerations over (a) the Yangtze River Middle Reaches, (b) Beijing–Tianjin–Hebei, (c) Pearl River Delta, and (d) Yangtze River Delta (d), which is superimposed with elevation."

**Response:**

Done! Thank you for your good suggestions. **(Lines 166-170)**

[7]. Fig. 3's y-axis for PEI lacks clarity—specify that "-" indicates dimensionless (after scaling by 1000). Also, "across the FOUR regions and seasons." is not correct, I only saw three seasons corresponding to three rows of line plots.

**Response:**

Sorry for the careless. Done! Thank you. And the sentence has been modified to: **"Average point line plots of nsRR, STH, LWP, IWP, and PEI under three AOD conditions for convective precipitation across the four regions and three seasons." (Lines 342-343)**

[8].Figure 5's title for X-axis lacks clarity—specify that "-" indicates dimensionless or directly delete "-" (as we all know AOD is dimensionless).

**Response:**

Done! Thank you. We have specified that **"-" indicates dimensionless.**

[9].Figures 2-3, 5-7: Spr. Sum. and Aut. are not standard abbreviation for three different seasons.

**Response:**

Thank you. We have replaced "Spr.", "Sum.", "Aut." with names: **"Spring", "Summer", "Autumn"** in all figures (Figs. 2-9; Figs .S9,11,13-22).

[10]."Conclusion" -> "Conclusions"

**Response:**

Done! Thank you.

We sincerely thank you for their insightful and constructive feedback, which has been invaluable in improving the quality and clarity of our manuscript. The revisions made in response to these comments have significantly strengthened the paper's methodological transparency, analytical depth, and conceptual framing. We believe that the revised manuscript now presents a more robust and compelling analysis of aerosol-precipitation interactions across China's megacity clusters. We look forward to your further feedback.